# INPP4B protects from metabolic syndrome and associated disorders

Manqi Zhang[1], Yasemin Ceyhan[2], Elena M. Kaftanovskaya[2], Judy L. Vasquez[2], Jean Vacher[3], Filip K. Knop [4,5,6,7], Lubov Nathanson[8], Alexander I. Agoulnik [2,9,10], Michael M. Ittmann [11,12] & Irina U. Agoulnik [2,10,13✉]

A high fat diet and obesity have been linked to the development of metabolic dysfunction and the promotion of multiple cancers. The causative cellular signals are multifactorial and not yet completely understood. In this report, we show that Inositol Polyphosphate-4-Phosphatase Type II B (INPP4B) signaling protects mice from diet-induced metabolic dysfunction. INPP4B suppresses AKT and PKC signaling in the liver thereby improving insulin sensitivity. INPP4B loss results in the proteolytic cleavage and activation of a key regulator in de novo lipogenesis and lipid storage, SREBP1. In mice fed with the high fat diet, SREBP1 increases expression and activity of PPARG and other lipogenic pathways, leading to obesity and non-alcoholic fatty liver disease (NAFLD). *Inpp4b*−/− male mice have reduced energy expenditure and respiratory exchange ratio leading to increased adiposity and insulin resistance. When treated with high fat diet, *Inpp4b*−/− males develop type II diabetes and inflammation of adipose tissue and prostate. In turn, inflammation drives the development of high-grade prostatic intraepithelial neoplasia (PIN). Thus, INPP4B plays a crucial role in maintenance of overall metabolic health and protects from prostate neoplasms associated with metabolic dysfunction.

[1] Department of Medicine, Duke University, Durham, NC, USA. [2] Department of Human and Molecular Genetics, Herbert Wertheim College of Medicine, Florida International University, Miami, FL, USA. [3] Department of Medicine, Institut de Recherches Cliniques de Montréal, Université de Montréal, Montréal, QC, Canada. [4] Center for Clinical Metabolic Research, Gentofte Hospital, University of Copenhagen, Hellerup, Denmark. [5] Department of Clinical Medicine, Faculty of Health and Medical Sciences, University of Copenhagen, Copenhagen, Denmark. [6] Novo Nordisk Foundation Center for Basic Metabolic Research, Faculty of Health and Medical Sciences, University of Copenhagen, Copenhagen, Denmark. [7] Steno Diabetes Center Copenhagen, Gentofte, Denmark. [8] Institute for Neuro Immune Medicine, Dr. Kiran C. Patel College of Osteopathic Medicine, Nova Southeastern University, Ft. Lauderdale, FL, USA. [9] Department of Obstetrics and Gynecology, Baylor College of Medicine, Houston, TX, USA. [10] Biomolecular Sciences Institute, Florida International University, Miami, FL, USA. [11] Department of Pathology and Immunology, Baylor College of Medicine, Houston, TX, USA. [12] Michael E. DeBakey Department of Veterans Affairs Medical Center, Houston, TX, USA. [13] Department of Molecular and Cellular Biology, Baylor College of Medicine, Houston, TX, USA. ✉email: iagoulni@fiu.edu

n 2017, overconsumption and unhealthy diets contributed to 11 million deaths globally. Diet-associated deaths from cancers and diabetes were ranked second and third behind cardiovascular disease[1]. The National Center for Health Statistics reported in 2016 that over 80% of adult males aged 20 years and older in the United States were overweight or obese. Among this age group, 70% of obese men and 30% of normal weight men will develop metabolic dysfunction[2]. The consumption of a high-fat diet (HFD) is recognized as a leading cause of obesity, metabolic dysfunction, elevated chronic inflammation[3], and cancer mortality[4]. Major indicators of metabolic dysfunction include elevated glucose levels, insulin resistance, and the expansion of mesenteric and omental adipose tissues[5]. Importantly for men, HFD and resulting obesity are tightly associated with lower urinary tract syndromes, including benign prostatic hyperplasia (BPH)[6], accelerated progression of prostate cancer, and decreased prostate cancer patient survival rates[7,8].

Increasing evidence suggests that non-alcoholic fatty liver disease (NAFLD) is the best clinical indicator of metabolic dysfunction in both obese and lean individuals[9,10]. Hepatic lipid storage occurs as a result of dysregulation in the insulin signaling pathway. Activation of the insulin receptor in liver, muscle, and fat cells leads to the activation of PI3K/AKT signaling pathways[11]. High levels of circulating diacylglycerol (DAG) lipids that are associated with HFDs activate PKCs, which, in turn, phosphorylate and inhibit the insulin receptor, lowering the insulin sensitivity[12]. In both mice and men, obesity elevates DAG levels and activates PKC signaling in multiple tissues, leading to the development of NAFLD and insulin resistance. Clinically, the levels of circulating DAG and PKCε activation are the strongest predictors of insulin resistance in obese men[13].

Importantly, a HFD is not exclusively responsible for metabolic syndrome as it has been reported that some obese individuals maintain a healthy metabolic profile[12,14,15]. There are several allelic variants that may impact the effects of known determinants of metabolic health. Polymorphisms in peroxisome proliferator-activated receptor γ (PPARG), adiponectin (ADIPOQ), leptin receptor (LEPR), and insulin receptor substrate 2 (IRS2) have been linked to obesity and type 2 diabetes mellitus (T2D)[16–19]. However, the specific signaling pathways that are disrupted during the initial stages of metabolic dysfunction are still not known.

Inositol Polyphosphate-4-Phosphatase Type II B (INPP4B) dephosphorylates both membrane lipids and phosphoproteins[20]. We have shown that INPP4B suppresses the AKT and PKC pathways[20–22], which are the major mediators of metabolic dysfunction, suggesting a role for INPP4B in the regulation of metabolic health. We tested this hypothesis using $Inpp4b^{-/-}$ male mice to compare their metabolic fitness to that of the wild-type (WT) males. We found that INPP4B has diet-dependent and independent actions in the mouse liver, adipose tissue, pancreas, and prostate. When fed a low-fat diet (LFD), 3-month-old $Inpp4b^{-/-}$ males increased their fat-to-lean body mass ratio, developed liver steatosis and hyperglycemia. This metabolic dysfunction likely resulted from reduced ambulatory activity, energy expenditure, and respiratory exchange ratios. When fed a HFD, in addition to liver steatosis, $Inpp4b^{-/-}$ males also developed type II diabetes (T2D), inflammation of the visceral white adipose tissue (WAT) and prostate, and prostatic intraepithelial neoplasia (PIN). Thus, the loss of INPP4B drives metabolic dysfunction by increasing hepatic lipogenesis, elevating systemic and local inflammation, and promoting the neoplastic transformation of the prostate in obese males.

## Results
### Accelerated weight gain in $Inpp4b^{-/-}$ male mice fed with a high-fat diet. $Inpp4b$ expression was previously reported in

multiple tissues[23]. Analysis of mouse tissues showed that the highest expression of $Inpp4b$ is in the testis, mammary gland fat pad, epididymal white adipose tissue (eWAT), muscle, bladder, and the ventral and dorsolateral lobes of the prostate. Among distinct adipose tissues, highest level of $Inpp4b$ expression was observed in mesenteric WAT (mWAT) and lowest in brown adipose tissue (BAT). Lower levels of $Inpp4b$ mRNA were also detected in the pancreas and liver (Fig. 1a, b), indicating that $Inpp4b$ is expressed in all major organs involved in insulin resistance. $Inpp4b^{-/-}$ males fed with a HFD gained significantly more weight than age-matched WT controls (Fig. 1c). At 3 months of age, obesity in $Inpp4b^{-/-}$ males was not accompanied by changes in body length, blood pressure, or heart rate (Supplementary Fig. 1a–c). A comparison of two representative fat depots, visceral WAT, the #4 mammary fat pad, and eWAT, revealed a significant weight increase in the HFD-fed $Inpp4b^{-/-}$ males compared to the WT males (Supplementary Fig. 1d, e). Total body composition analysis by QMR revealed that knockout males have an increased fat-to-lean mass ratio (Fig. 1d–f). While overall body weights of experimental animals did not change in the course of CLAMS analysis, total food and water intake as well as the home cage activities were significantly decreased in knockout mice (Fig. 1g–i). Consistently, energy expenditure and respiratory exchange ratios were significantly decreased in $Inpp4b^{-/-}$ males during light cycle (Fig. 1j, k and Supplementary Fig. 1f, g). Histological comparison of the inguinal, retroperitoneal, mesenteric and eWAT, as well as BAT, did not reveal visible differences between 4 experimental groups (Supplementary Fig. 2).

**Metabolic and inflammatory changes in adipose tissue of $Inpp4b^{-/-}$ males.** In eWAT, neither diet nor $Inpp4b$ knockout altered the protein levels of HK2, an enzyme that mediates the commitment step of glycolysis (Fig. 2a and Supplementary Fig. 3a). Expression of fatty acid synthase (Fasn) mRNA was increased with HFD (Supplementary Fig. 3b) and the FASN protein levels were further increased in eWAT of HFD-fed $Inpp4b^{-/-}$ mice (Fig. 2a, b), suggesting a role for INPP4B in lipid accumulation. Expression of $Inpp4b$ was not affected in eWAT tissue by the HFD (Supplementary Fig. 3c). Thus, the loss of INPP4B combined with a HFD-activated lipogenesis led to a significant adipose expansion.

WAT secretes a broad variety of adipokines and cytokines that modulate inflammation and insulin signaling[24–27]. We tested whether INPP4B regulates the inflammatory status of adipose tissue in HFD-fed males. The eWAT of HFD-fed $Inpp4b^{-/-}$ mice had a significantly increased ratio of leptin-to-adiponectin expression (Fig. 2c) and elevated expression of the inflammatory cytokines $Il6$ and $Tnf$ (Fig. 2d, e). The adipose tissue in HFD-fed knockout males showed an increased expression of the macrophage markers $Adgre1$ and $CD68$ (Fig. 2f, g). The highest protein levels of CD68 were observed in the eWAT of the HFD $Inpp4b^{-/-}$ group, suggesting that the INPP4B deficiency promotes the inflammatory response of adipose tissue in diet-induced obesity (Fig. 2a, h). As shown in Fig. 2d–h, both HFD and INPP4B loss contributed to the increases in inflammatory cytokines and infiltrating macrophage markers in the adipose tissue.

Using adipokine arrays, we compared the protein levels of 40 adipokines from the eWAT of WT and $Inpp4b^{-/-}$ mice fed with LFD or HFD (Supplementary Fig. 3d). The array analysis confirmed that compared to LFD WT males, adipose tissue leptin/adiponectin protein-level ratios were increased in LFD $Inpp4b^{-/-}$, HFD WT, and HFD $Inpp4b^{-/-}$ groups. However, the differences between these 3 groups did not reach statistical significance (Supplementary Fig. 3d). Consistent with the increased presence of CD68+ cells, the eWAT from $Inpp4b^{-/-}$ mice expressed significantly higher levels of the proinflammatory chemokine, monocyte chemoattractant protein-1 (MCP-1), which promotes macrophage infiltration and

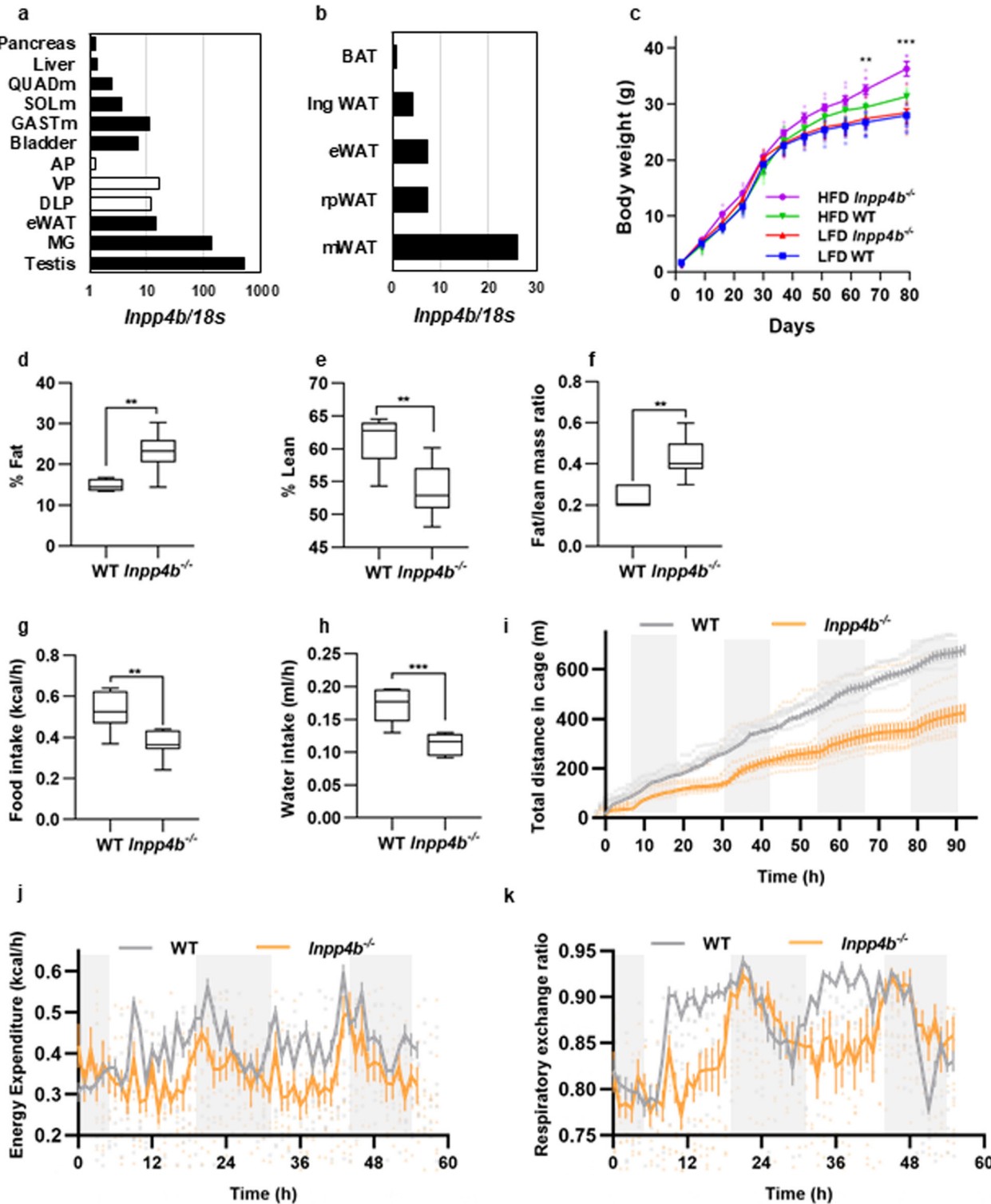

**Fig. 1 Inpp4b−/− males accumulate more fat due to decreased activity, energy expenditure, and respiratory exchange ratio. a** Expression of *Inpp4b* in mouse tissues. **b** Expression of *Inpp4b* in epididymal, retroperitoneal, mesenteric, inguinal, and brown adipose tissue of FVB males. **c** Body weights of LFD WT ($N = 14$), LFD *Inpp4b−/−* ($N = 12$), HFD WT ($N = 9$), and HFD *Inpp4b−/−* ($N = 10$) mice were recorded every week beginning on post-partum day 2. LFD WT, LFD *Inpp4b−/−*, HFD WT, and HFD *Inpp4b−/−* groups are represented by the blue line with squares, red line with up-pointing triangles, green line with down-pointing triangles, and purple line with circles, respectively (**$p = 0.0048$, ***$p = 0.001$). **d-f** Percent fat (**$p = 0.0012$) (**d**), and percent lean (**$p = 0.0014$) (**e**) were determined by NMR and the ratios of fat-to-lean mass (**$p = 0.00015$) (**f**) were calculated for WT ($N = 6$) or *Inpp4b−/−* ($N = 10$) mice. **g, h** Hourly food (**$p = 0.0027$) and water intake (***$p = 0.00023$) were recorded for the WT ($N = 6$) or *Inpp4b−/−* ($N = 8$) mice. **i** Mouse ambulatory activity was detected with XYZ beam array over 92 h in WT ($N = 6$) or *Inpp4b−/−* ($N = 10$) mice. Total distances were acquired by MetaScreen and processed using ExpeData. **j, k** Animals were acclimated in cages for the first 42 h. Time course of energy expenditure (**j**) and respiratory exchange ratio (**k**) over subsequent 55 h in WT ($N = 6$) or *Inpp4b−/−* ($N = 8$) male mice using CLAMS analysis.

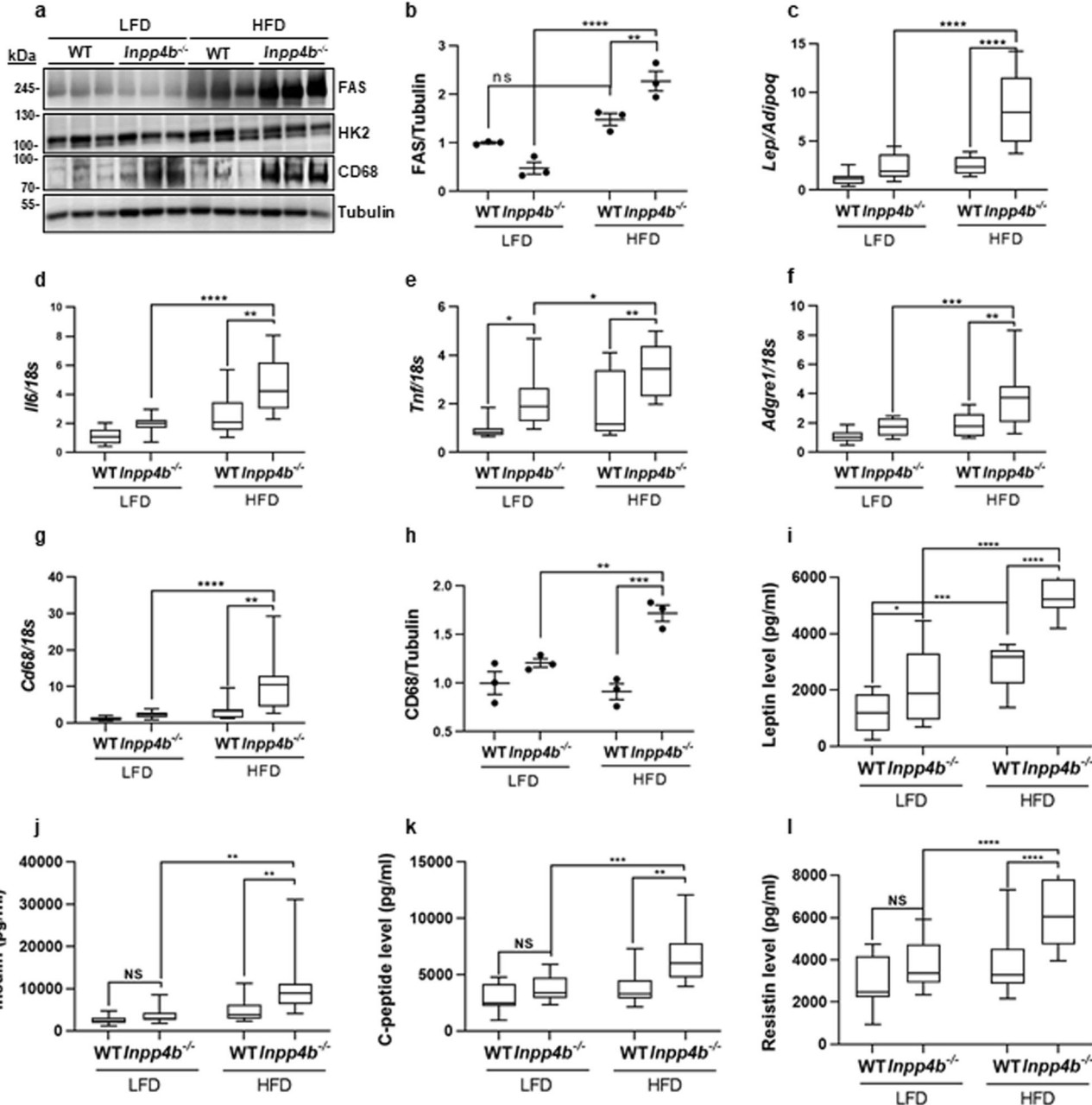

**Fig. 2 Comparison of the metabolic and inflammatory profiles of adipose tissues from WT and *Inpp4b*[−/−] males. a** Protein lysates from eWAT were analyzed for FAS, HK2, CD68, and tubulin by western blot. **b** Quantification of FAS protein expression in (**a**). The FAS values were normalized to tubulin and are shown as fold change (N = 3 per group). **c–g** RNA was extracted from mouse eWAT of LFD WT (N = 12), LFD *Inpp4b*[−/−] (N = 11), HFD WT (N = 8), and HFD *Inpp4b*[−/−] (N = 9) mice. The expression levels of *Lep/Adipoq* (**c**), *Il6* (**d**), *Tnf* (**e**), *Adgre1* (**f**), and *Cd68* (**g**) were assayed by qRT-PCR using *18S* as an internal control. **h** Quantification of CD68 protein in (**a**). Average expression in LFD WT mice was designated as 1 (N = 3 per group). **i–l** Concentrations of leptin (**i**), insulin (**j**), C-peptide (**k**), Resistin (**l**) in serum of LFD WT (N = 11), LFD *Inpp4b*[−/−] (N = 11), HFD WT (N = 9), and HFD *Inpp4b*[−/−] (N = 9) mice (*p < 0.05, **p < 0.01, ***p < 0.001, ****p < 0.0001. Exact p values are provided in Supplementary Data 2).

is known to be increased in obese diabetic mice[28,29]. Adipose expansion and inflammation are downregulated by INPP4B in obese animals, suggesting a protective role for INPP4B from metabolic syndrome. Analysis of metabolic hormones in the serum showed significant increases in leptin, insulin, C-peptide, and resistin in HFD *Inpp4b*[−/−] males compared to WT on either diet and LFD *Inpp4b*[−/−] mice (Fig. 2i–l and Supplementary Fig. 3e).

**Inpp4b-deficient mice develop hyperglycemia on low-fat diet and type 2 diabetes on high-fat diet.** The expansion and increased inflammation of adipose tissues suggested that

*Inpp4b*[−/−] animals are insulin resistant. In an independently conducted QTL analysis in mice, Mu et al. have shown an association between the *D8Mit195* locus, mapped immediately next to the *Inpp4b* gene, and elevated plasma glucose levels (LRS = 10.4, LOD = 2.3)[30]. The increase in the *Lep/Adipoq* ratio observed in HFD *Inpp4b*[−/−] group is characteristic to obese diabetic mice and men[31,32]. Glucose tolerance tests revealed that WT HFD-fed mice had an elevated fasting blood glucose level compared to the lean WT males. Upon administration of a bolus of oral glucose to WT males, serum glucose levels rose higher in HFD group than in LFD but recovered to normal after 120 min.

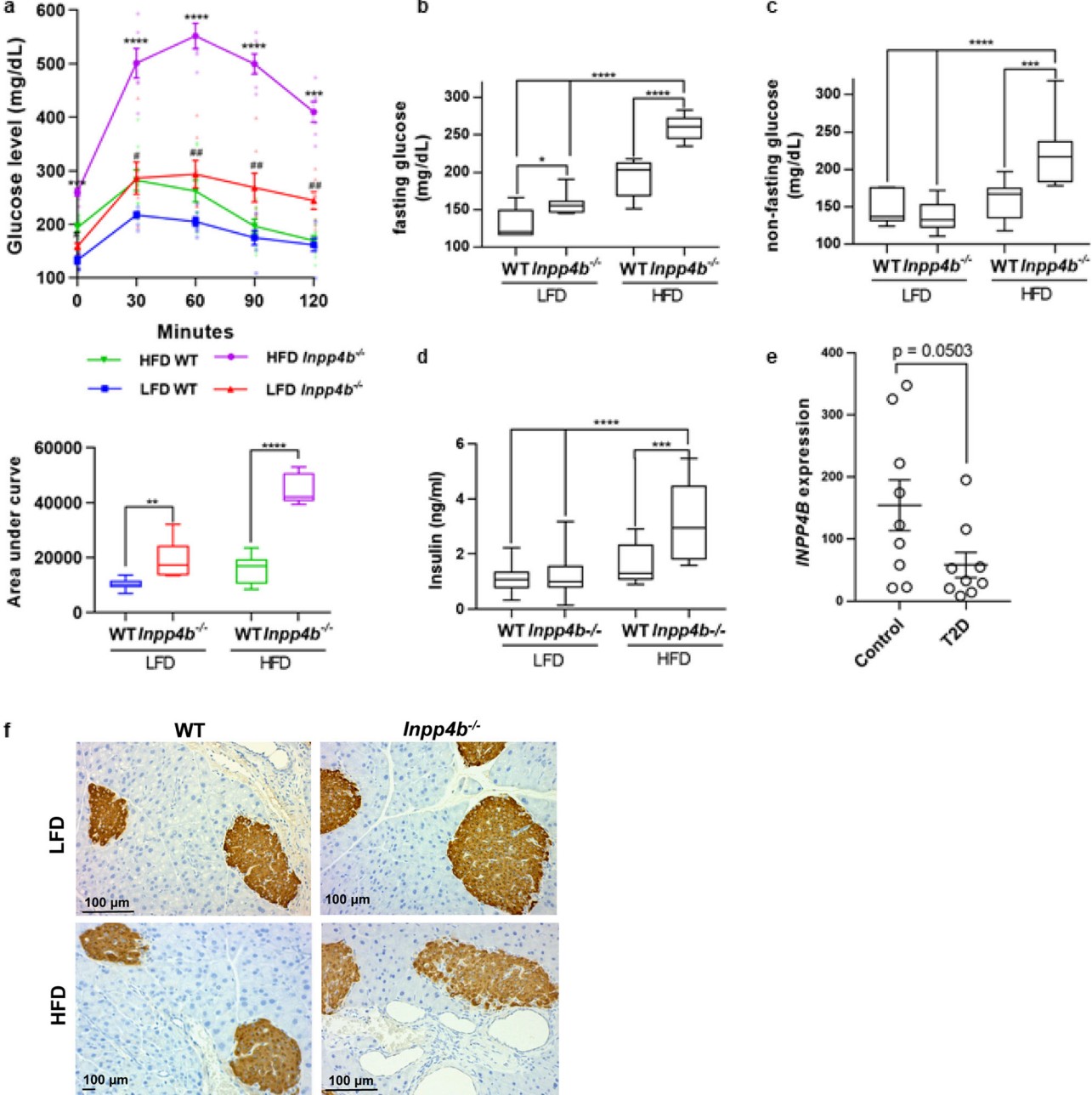

**Fig. 3 INPP4B protects mice from hyperglycemia and diabetes. a** Oral glucose tolerance test of LFD WT ($N = 7$), LFD $Inpp4b^{-/-}$ ($N = 7$), HFD WT ($N = 9$), and HFD $Inpp4b^{-/-}$ ($N = 6$) mice following a 2 g/kg oral glucose challenge. **#** is used to indicate statistically significant difference between LFD groups and **\*** is used to indicate statistically significant difference for HFD groups. LFD WT, LFD $Inpp4b^{-/-}$, HFD WT, and HFD $Inpp4b^{-/-}$ groups are represented by blue line with squares, red line with up-pointing triangles, green line with down-pointing triangles, and purple line with circles, respectively. **b, c** Blood glucose was measured in the same group as (**a**) after 6 h of fasting (**b**) or non-fasting (**c**) LFD WT, LFD $Inpp4b^{-/-}$, HFD WT and HFD $Inpp4b^{-/-}$ mice. **d** Blood insulin was measured by ELISA in the same groups as (**a**). **e** INPP4B expression in livers of subjects with or without type II diabetes (T2D) were exported from GSE15653. **f** Representative pancreata sections from LFD WT, LFD $Inpp4b^{-/-}$, HFD WT, and HFD $Inpp4b^{-/-}$ mice were stained for insulin and counterstained with hematoxylin ($^*p < 0.05$, $^{***}p < 0.001$, $^{****}p < 0.0001$. Exact $p$ values are provided in Supplementary Data 2).

When fed with either LFD or HFD, $Inpp4b^{-/-}$ mice had elevated glucose levels that they were unable to fully clear after 120 min (Fig. 3a, b). Insulin tolerance testing revealed that $Inpp4b^{-/-}$ males were resistant to insulin treatment (Supplementary Fig. 4a). The rapid increase of glucose to nearly 600 mg/dl during the glucose tolerance test, insulin resistance, and the inability to clear blood glucose in obese $Inpp4b^{-/-}$ males were consistent with obesity-induced T2D[33]. Circulating, non-fasting glucose and insulin levels were also significantly elevated in HFD-fed mutant animals compared to every other group (Fig. 3c, d). Despite normal levels of circulating insulin in LFD $Inpp4b^{-/-}$ males, the rate of hepatic gluconeogenesis was increased, confirming that lean knockout males developed hyperglycemia (Supplementary Fig. 4b). Similar to our HFD $Inpp4b^{-/-}$ male mice, gene expression analysis in the livers of non-diabetic ($N = 9$) and T2D obese individuals ($N = 9$) (GSE15653) revealed that INPP4B expression was decreased in the obese, diabetic group (Fig. 3e)[34]. As expected, pancreatic islets were positive for insulin staining by

IHC (Fig. 3f). Consistent with the increase in blood insulin levels, the expression of proinsulin convertase, *Pcsk1*, was increased in the pancreata of LFD-fed *Inpp4b*[−/−] mice and in both HFD-fed groups (Supplementary Fig. 4c). Both HFD and *Inpp4b* knockout decreased expression of the peptide hormone cholecystokinin (*Cck*), an appetite suppressant and stimulant of pancreatic enzyme secretion[35] (Supplementary Fig. 4d). The markers of pancreatic β-cell viability and inflammation, *Furin* and *Il6*, and the unfolded protein response (UPR), *Mafa*, did not significantly change (Supplementary Fig. 4e–g) at 3 months of age. Thus, our results and results from other groups suggest that INPP4B is a metabolic modulator that protects from diet-induced T2D.

**Loss of INPP4B leads to NAFLD**. NAFLD is the best predictor of insulin resistance[9]. To determine whether insulin resistance in *Inpp4b*[−/−] mice is associated with NAFLD, we first compared the liver weights in WT and *Inpp4b*[−/−] males fed with LFD or HFD. Strikingly, the liver weights were significantly elevated in *Inpp4b*[−/−] knockouts independent of the diet (Fig. 4a). Morphometric quantification of H&E stained slides revealed severe steatosis in the livers of all *Inpp4b*[−/−] mice (Fig. 4b, c). Livers in knockout animals featured all major histological characteristics of NAFLD, such as accumulation of fat (Fig. 4d), microvesicular and macrovesicular steatosis, hepatocellular hypertrophy, pyknotic nuclei, and Mallory bodies (Fig. 4e–g)[36]. Liver triglyceride amounts were significantly increased in *Inpp4b*[−/−] mice compared to WT controls in both LFD and HFD groups (Supplementary Fig. 5a). In the livers of WT male mice, the HFD caused modest increases in the expression of *Inpp4b* and *Cd68*, a marker of infiltrating macrophages (Supplementary Fig. 5b, c). In the human liver, INPP4B was detected in the cytoplasm and cell membrane of normal hepatocytes and its protein levels decreased in primary and, especially, metastatic hepatocellular carcinoma[37], which undergo significant metabolic reprogramming. Similar to the mouse liver, hepatic *INPP4B* expression in patients with NAFLD was significantly reduced compared to healthy controls (GSE126848)[38] and there was a highly significant correlation between steatosis score and *INPP4B* mRNA levels (Fig. 4h, i). Thus, the loss of INPP4B in mice is similar to the decline in *INPP4B* expression observed in steatotic livers of NAFLD patients[38].

**NAFLD in *Inpp4b*[−/−] males is caused by increased lipogenesis, WAT inflammation, and activation of AKT and PKC signaling**. To determine the molecular pathways that protect WT mice on HFD from NAFLD, we performed RNA-seq analysis of liver samples from WT and *Inpp4b*[−/−] males on a HFD. Analysis of differentially expressed genes (DEG) with DAVID functional annotation module and Gene Set Enrichment Analysis (GSEA) revealed that *Inpp4b* deletion affected lipid and glucose metabolisms, drug detoxification, peroxisome biogenesis, and expression of the T2D-associated genes (Fig. 5a, b and Supplementary Data 2). Lipid metabolism and peroxisome biogenesis are known targets of the PPAR family of transcription factors[37,39,40]. PPARG is a transcriptional regulator of fatty acid uptake and storage, while PPARA activity stimulates fatty acid oxidation[41]. Expression of *Pparg* increased in both LFD- and HFD-fed *Inpp4b*[−/−] males compared to WT controls (Fig. 5c). Correspondingly, expression of PPARG target genes regulating lipid delivery, internalization, triglyceride synthesis, and fatty acid metabolism increased in *Inpp4b*[−/−] livers (Fig. 5d–g). Strikingly, a group of genes involved in sexually dimorphic drug and lipid metabolism, *Cyp2b9*, *Cyp2b13*, and *Hao2* (GSE 89091)[42], were strongly downregulated by INPP4B as their expression dramatically increased in the livers of knockout males (Fig. 5g–i). *Ppara* expression was modestly increased in LFD *Inpp4b*[−/−] group only (Supplementary Fig. 5d), potentially contributing to partial NAFLD resistance in lean

knockout animals. Similar to *Inpp4b*[−/−] mice, liver samples collected in the course of Roux-en-Y gastric bypass surgery from 1008 obese patients (GSE24335) exhibited a highly significant negative correlation between *INPP4B* and *PPARG* expression[43] (Fig. 5j). Thus, expression data in mice and men support our finding that INPP4B regulates PPARG levels in the liver.

Disparate spatial localization of these proteins suggests the existence of an intermediate regulatory protein. SREBP1 is a transcriptional activator of many lipogenic enzymes, including *Pparg*[44,45]. Based on our RNA-seq analysis, the levels of *Srebf1* mRNA did not significantly increase in *Inpp4b*[−/−] livers (Fig. 6a). However, activation of SREBP1 requires proteolytic cleavage and nuclear translocation[46,47]. Comparison of the precursor (125 kDa) and transcriptionally active (55 kDa) forms of SREBP1 showed a significant increase in both proteins' levels due to *Inpp4b* knockout and HFD treatment (Fig. 6b, c). We and others have previously reported that INPP4B regulates Akt and PKC[21,22,48] pathways. Consistently, phosphorylation of AKT, PKCβII, and PKCζ was increased in the livers of HFD-fed and knockout animals with the highest expression in HFD *Inpp4b*[−/−] group (Fig. 6d–f). Interestingly, phosphorylation of PKCε remained unchanged in all groups. Conversely, 16-h treatment of *Inpp4b*[−/−] males with the Akt inhibitor, AZD5363, and pan-PKC inhibitor, BIM-I, led to a significant decline in both the precursor and mature forms of SREBP1 (Fig. 6g, h). Importantly, the expression of primary SREBP1 target genes, *Mogat1*, *Cyp2b13*, and *Cyp2b9*[49], were significantly decreased in the livers of mice treated with the inhibitors BIM-I and AZD5363 confirming direct involvement of Akt and PKC in the regulation of SREBP1 signaling (Fig. 6i).

Fasting suppresses lipogenesis by reducing the processing, and therefore activity, of the SREBP1 transcription factor. Remarkably, we found that downregulation of Akt signaling and SREBP1 processing during fasting was abolished in *Inpp4b*[−/−] males (Fig. 6j–l). There is strong evidence that SREBP1 expression and proteolysis in the liver is induced by insulin, Akt, retinoids, PKC, and its own activated form[46,47,50–55]. Thus, the activation of insulin, retinol, Akt, and PKC signaling pathways in obese knockout males contributes to the proteolytic cleavage of SREBP1 in the livers of HFD *Inpp4b*[−/−]. The cleaved form of SREBP1 is responsible for activating the expression of *Pparg* and other lipogenic enzymes that lead to the development of NAFLD (Figs. 2a, 3, 4, and 5b–f). The inability to cease de novo lipogenesis (DNL) during fasting cycles likely stimulates hepatic accumulation of lipids in *Inpp4b*[−/−] males.

**HFD causes neoplastic changes in prostates of *Inpp4b*[−/−] mice**. We have previously reported that *Inpp4b*[−/−] mice on regular chow have the same weight as WT animals and that they do not develop tumors in the first 4 months of age, while some hyperplastic changes developed in 1-year-old animals[56]. The increased leptin/adiponectin ratios and adipose inflammation are risk factors for the development of prostate cancer in men[57,58] and were observed in HFD *Inpp4b*[−/−] males. We tested whether metabolic dysfunction triggered by the HFD would lead to an earlier and more severe prostate phenotype in knockout male mice. Indeed, all *Inpp4b*[−/−] males ($N = 11$) developed high-grade PIN by 11 weeks of age, while none of the WT males ($N = 8$) on the same diet did. Anterior prostate (AP), dorsolateral prostate (DLP), and ventral prostate (VP) lobes exhibited various degrees of epithelial expansion and loss of polarity. The most extensive histological changes were observed in DLP glands, including epithelial and basal layer expansion, piling up of epithelial cells (Fig. 7a, b), loss of polarity (Supplementary Fig. 6a), a discontinuous fibromuscular layer around individual glands (Supplementary Fig. 6b), and nuclear atypia (Supplementary Fig. 6c).

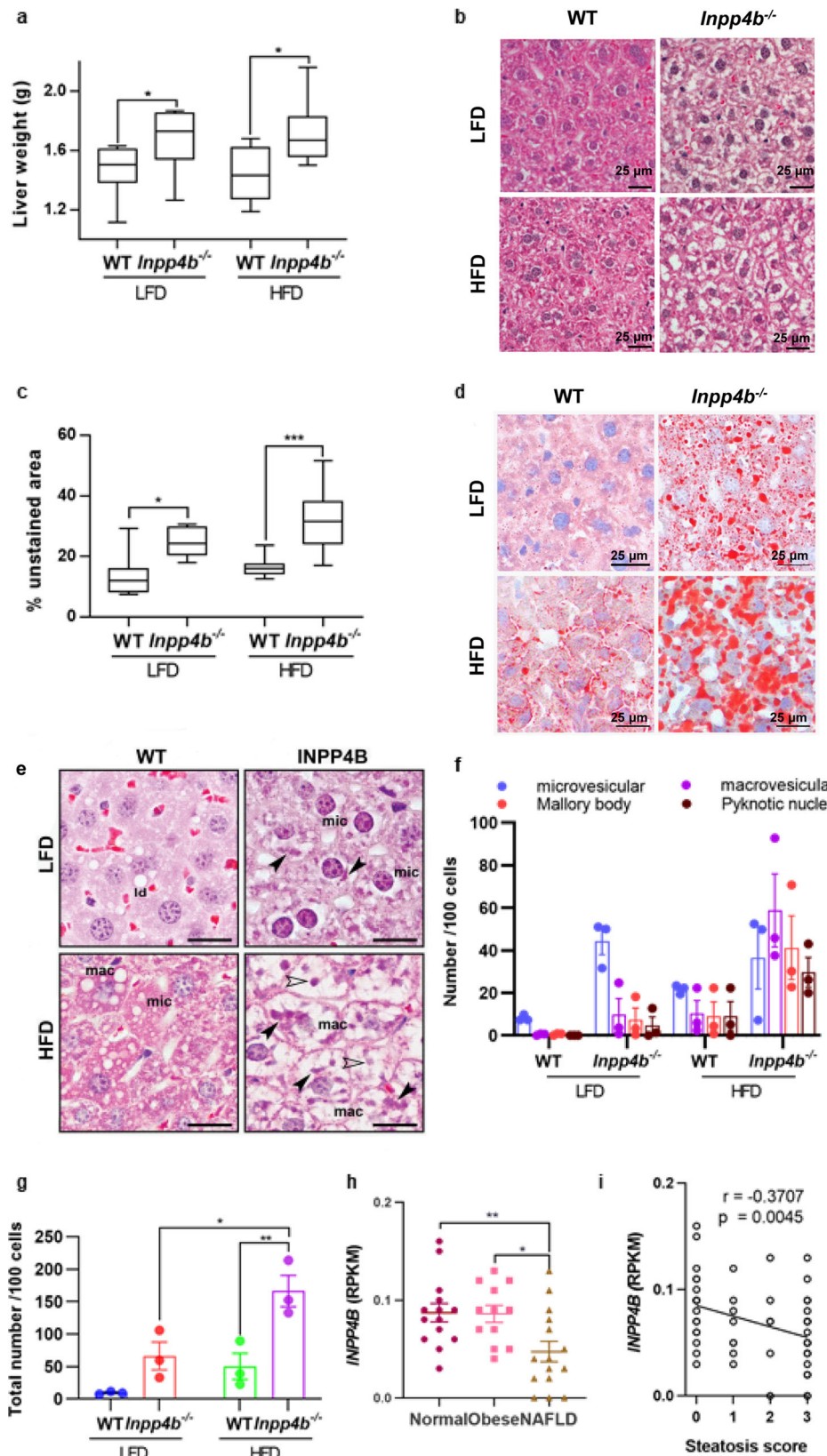

Androgen receptor activity, cell signaling, and inflammation are among the factors most strongly contributing to neoplastic transformation of the prostate[59]. The protein and mRNA levels of AR and PTEN were not significantly altered by INPP4B loss on either diet (Supplementary Fig. 7a–d). While the HFD led to increased macrophage infiltration in the prostates of both WT and mutant males, there was no difference between the two genotypes in *Cd68* expression (Supplementary Fig. 7e). Inflammatory infiltrate was observed in the prostate stromata of obese WT and knockout males (Supplementary Fig. 7f, marked with *). Staining with the Ki67 antibody indicated increased cellular proliferation in the prostates of HFD *Inpp4b*$^{-/-}$ males

**Fig. 4 INPP4B protects mice from liver steatosis. a** Livers were dissected from LFD WT ($N = 11$), LFD $Inpp4b^{-/-}$ ($N = 13$), HFD WT ($N = 9$), and HFD $Inpp4b^{-/-}$ ($N = 10$) mice. Liver weights in 4 experimental groups (\*$p = 0.0422$ for LFD groups and \*$p = 0.0374$ for HFD groups). **b** Representative H&E staining of liver sections from designated experimental groups used for steatosis analysis. **c** Morphometric quantification of the unstained area on H&E stained liver sections (\*$p = 0.111$, \*\*\*$p = 0.0004$). **d** Representative images of the Oil Red O/hematoxylin staining of frozen liver sections in designated groups. **e** Characteristic features of hepatocellular steatosis in livers of HFD and LFD $Inpp4b^{-/-}$ males. Examples of microvesicular (mic) and macrovesicular (mac) steatoses, Mallory bodies (closed arrowhead) and pyknotic nuclei (open arrowhead). **f** Number of individual features per 100 cells. **g** Total number of steatotic features per 100 cells in each treatment group (\*$p = 0.0118$, \*\*$p = 0.0052$). **h** INPP4B expression in Reads Per Kilobase of transcript, per Million mapped reads (RPKM) in normal liver tissue ($N = 14$), obese ($N = 12$ patients), and NAFLD ($N = 15$ patients) in GSE126848 dataset (exact $p$ values are provided in the Source data). One-way ANOVA (Dunnett's multiple comparison test) was used. **i** Correlation between INPP4B expression with the steatosis score in patients with NAFLD or NASH. Data were exported from GSE126848.

(Supplementary Fig. 8a, b). While AR protein levels did not change, its transcriptional activity was affected. Expression of AR-induced target genes, *Pbsn*, *Apof*, *Nkx3.1*, and *Msmb*, were significantly decreased in the DLP of LFD $Inpp4b^{-/-}$ males (Fig. 7c–f). HFD further decreased expression of AR target genes *Nkx3.1* and *Msmb* (Fig. 7e, f). However, we observed no change in the ability of AR to repress transcription of its target gene *Igfbp3* (Fig. 7g). Interestingly, HFD decreased the levels of *Inpp4b* expression in DLP by almost two-fold (Supplementary Fig. 8c). Levels of pAkt were increased in prostates of LFD $Inpp4b^{-/-}$ mice; no further increase was detected in HFD-fed groups (Supplementary Fig. 8d, e). Similarly, levels of PKCβII and pPKCζ were slightly increased in DLP and VP lobes of LFD $Inpp4b^{-/-}$ males and treatment with HFD did not further increase the signal (data not shown).

**HFD leads to prostate inflammation in $Inpp4b^{-/-}$ males.** Inflammation plays a key role in the etiology of prostate cancer[60]. We tested whether the development of prostate neoplasms in the HFD $Inpp4b^{-/-}$ males was accompanied by inflammation. In the dorsolateral lobe of the prostates of obese males, *Il6* and *Tnf* mRNA levels were significantly increased in $Inpp4b^{-/-}$ mice when compared to WT (Fig. 7h, i). As expected, expression of the potent proinflammatory cytokine, *Il1b*, was undetectable in the prostates of LFD mice. Consumption of HFD induced the expression of *Il1b* in the prostates in both genotypes. In the DLP, the loss of INPP4B resulted in a significant increase in *Il1b* expression when compared to WT (Fig. 7j). Protein levels of the proinflammatory cytokine IL6 were significantly increased in the DLP of HFD-fed $Inpp4b^{-/-}$ mice (Fig. 7k). While there are anatomical differences between mice and humans, the mouse DLP is considered most similar to the human peripheral zone, the site of origin of most human prostate cancers[61]. It is also important to note that in the prostate, the dorsolateral region features high level of *Inpp4b* expression (Supplementary Fig. 8c), suggesting a physiological function for INPP4B protein in that lobe.

## Discussion

Metabolic dysfunction is the cause of significant morbidity and mortality due to diabetes, cardiovascular abnormalities, and an increased risk of cancer incidence and mortality. Though epidemiological links between increased calorie consumption, cancer, and diabetes are well established, less is known about the specific genes that protect some obese individuals from metabolic dysfunction and neoplasia. In this report, we show that INPP4B is a key regulator of metabolic health in lean male mice and protects obese mice from diabetes and prostate neoplasia.

INPP4B has been shown to play a tumor suppressor role in prostate, breast, and other cancers[22,48,62,63], however, $Inpp4b^{-/-}$ mice do not develop malignancies[64]. Our data demonstrated significant metabolic changes in $Inpp4b^{-/-}$ males at 3 months of age and show that HFD exacerbated these changes. $Inpp4b^{-/-}$

males had an elevated fat-to-lean body mass ratio despite a somewhat lower food intake. This was likely due to their reduced activity and associated decline in energy expenditure and respiratory exchange ratio (Fig. 1 and Supplementary Fig. 1). INPP4B-dependent regulation of insulin sensitivity and its downstream Akt and PKC pathways led to the increased circulating levels of insulin and accelerated weight gain in HFD $Inpp4b^{-/-}$ males. Hepatic Akt and PKC signaling pathways are required for insulin sensitivity[12], and, in fact, we observed inefficient glucose clearance and insulin resistance in both LFD and HFD $Inpp4b^{-/-}$ animals (Fig. 3 and Supplementary Fig. 4a). $Inpp4b^{-/-}$ mice developed hyperglycemia on an LFD and T2D on a HFD. Insulin promotes hepatic lipogenesis largely through upregulation of SREBP1, a transcription factor that potently activates lipogenic pathways[65]. Taniguchi et al. reported that in mouse liver, Akt mediates glucose homeostasis and insulin tolerance while PKCζ stimulates expression of SREBP1[53,66]. Akt and PKC signaling also induce proteolytic cleavage of the SREBP1 precursor protein into a mature transcription factor[46,47] which regulates the expression and activity of PPARG[67,68]. Consistently, we found that INPP4B loss and a HFD cooperatively induced the levels of pAkt, pPKCβII, pPKCζ, and SREBP1 cleavage, leading to increased expression of *Pparg* and other lipogenic transcripts during fed state (Figs. 5 and 6). SREBP1 processing declines dramatically during fasting to inhibit lipogenesis in the liver. Our data suggest that fasting-induced inhibition of SREBP1 processing is regulated by INPP4B (Fig. 6j–l). As a direct result of these transcriptional and posttranslational changes in the expression and activation of crucial cellular signaling factors, we observed the development of hepatosteatosis, WAT expansion, and inflammation in $Inpp4b^{-/-}$ males which in turn were further exacerbated by a HFD. At 3 months of age, both $Inpp4b$ knockout and HFD increased circulating levels of the metabolic hormones insulin, C-peptide, leptin, and resistin (Fig. 2). Insulin and C-peptide are the products of the proteolytic cleavage of proinsulin. Increased leptin concentrations correlated with the increases in adipose tissue mass caused by HFD treatment and INPP4B loss (Figs. 1c and 2i). Increased resistin level is also consistent with our phenotype as this hormone mediates insulin resistance and links obesity to T2D in mice and men[69,70]. It is important to note that the $Inpp4b^{-/-}$ mouse model closely resembles human NAFLD. The downregulation of hepatic INPP4B expression in patients with T2D and advanced hepatosteatosis, as well as a highly significant negative correlation between the expression of INPP4B and PPARG was observed in livers of obese patients (Figs. 4g–i and 5j).

Metabolic dysfunction and adipose inflammation in HFD $Inpp4b^{-/-}$ males affected prostate gland development, as is often observed in men with this condition. HFD $Inpp4b^{-/-}$ males were significantly more susceptible to obesity-induced neoplastic transformation of the prostate than the WT males (Fig. 7a, b). Circulating leptin, local inflammation, and activated immune cells in prostate stroma likely contribute to the inflammatory

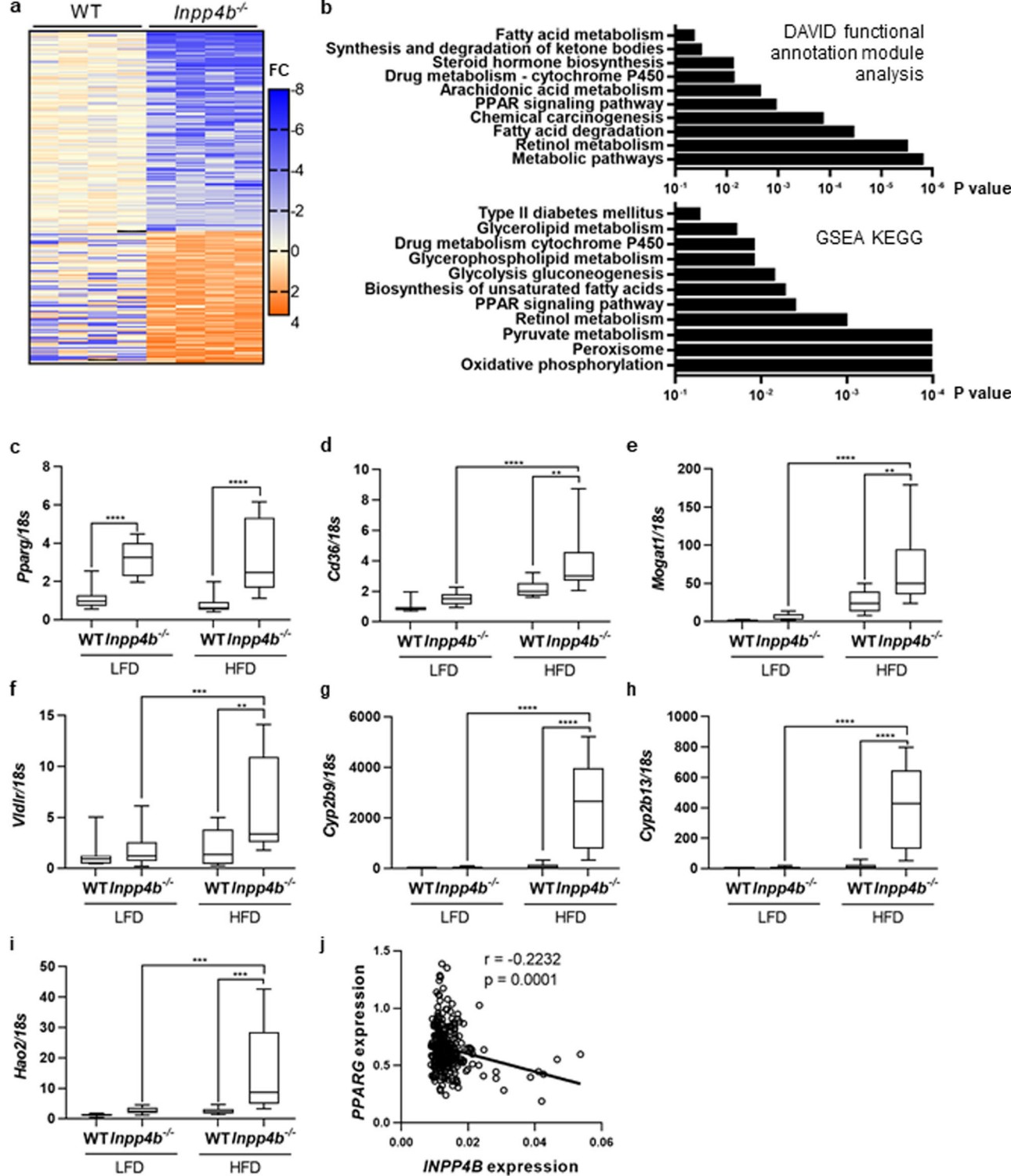

**Fig. 5 INPP4B regulates metabolic and PPAR pathways in mouse liver. a** RNAs were extracted from livers of HFD WT ($N = 4$) and HFD *Inpp4b*$^{-/-}$ ($N = 4$) mice and submitted for RNA sequencing. The changes in gene expression were compared between HFD WT and HFD *Inpp4b*$^{-/-}$ mice. **b** KEGG pathway functional enrichment analysis was done using DAVID functional annotation module or GSEA. The vertical axis represents the KEGG pathway terms significantly enriched by the loss of INPP4B in mouse liver. The horizontal axis shows the logarithmic scale of the *p* value. **c–i** RNA from LFD WT ($N = 10$), LFD *Inpp4b*$^{-/-}$ ($N = 12$), HFD WT ($N = 8$), and HFD *Inpp4b*$^{-/-}$ ($N = 10$) mice were analyzed for *Pparg* (**c**), *Cd36* (**d**), *Mogat1* (**e**), *Vldlr* (**f**), *Cyp2b9* (**g**), *Cyp2b13* (**h**), and *Hao2* (**i**) by qRT-PCR using *18S* as an internal control. **j** Correlation between *INPP4B* expression and *PPARG* expression in 289 human liver samples (GSE24335) (**p < 0.01, ***p < 0.001, ****p < 0.0001. Exact *p* values are provided in the Source data).

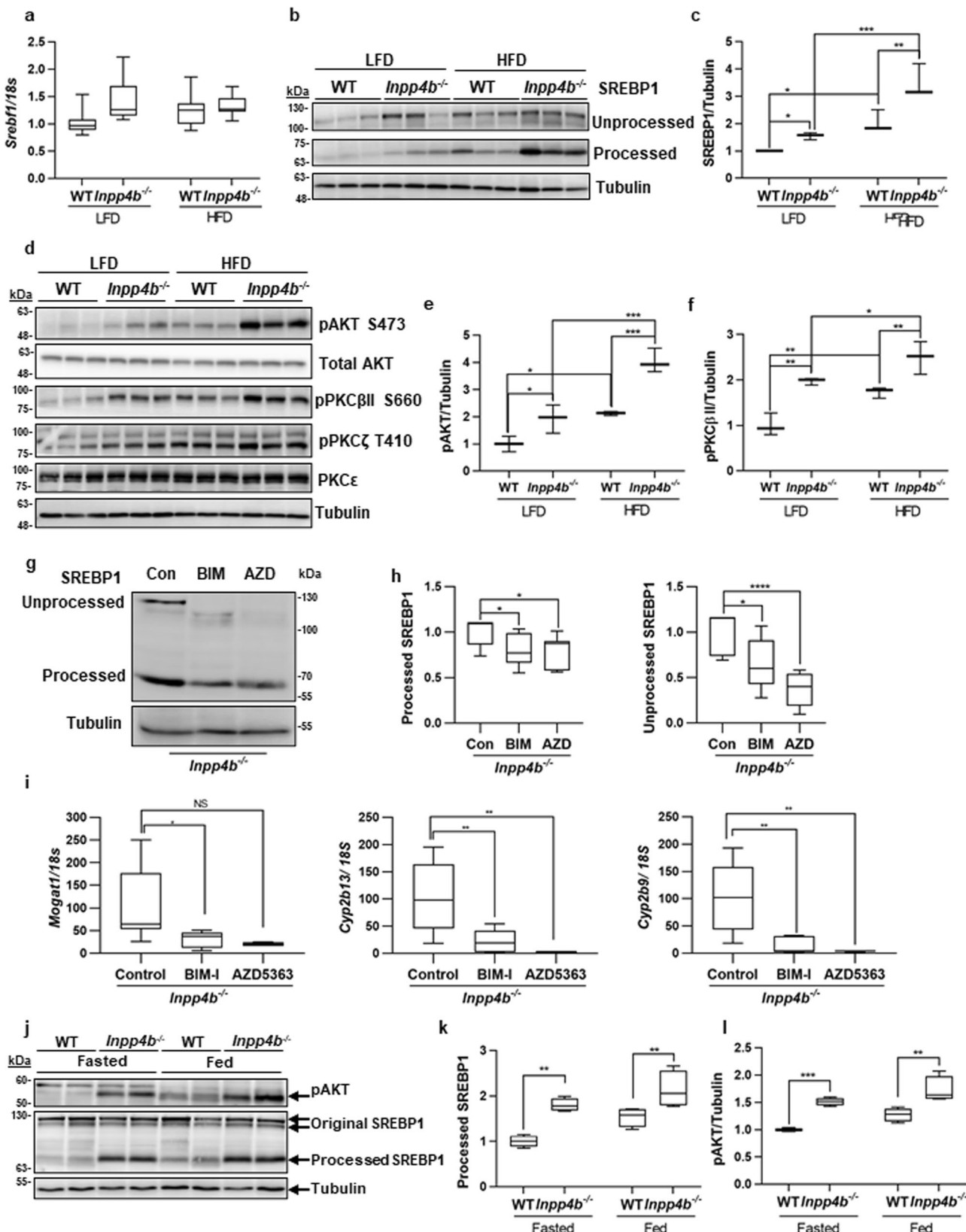

environment of the prostate[71–73]. At 3 months of age, only local inflammation was detected in the prostate (Fig. 7h–k). No significant changes between 4 experimental groups in serum levels of IL-6, TNFα, and IL-1β inflammatory cytokines were detected at 3 months of age. It was established that IL-6 is elevated in tissues of BPH and prostate carcinomas in men[74] and intratumoral IL-1β

levels significantly correlate with biochemical recurrence in prostate cancer patients[75]. Importantly, prostatic neoplasms were detected only in HFD *Inpp4b*−/− males, suggesting a modulatory role for INPP4B in obesity-induced prostatic neoplasms. We have shown that the prostatic glands displayed increased proliferation of the epithelial layer and occasional breach of prostate glands'

**Fig. 6 INPP4B regulates lipogenesis and AKT and PKC signaling pathways in mouse liver. a** RNA from Fig. 4(c–i) were analyzed for *Srebf1* expression by qRT-PCR using *18S* as control. **b** Proteins were extracted from livers of LFD WT, LFD *Inpp4b*$^{-/-}$, HFD WT, and HFD *Inpp4b*$^{-/-}$ mice and used to compare levels of unprocessed SREBP1, processed SREBP1, and tubulin by western blotting. **c** Quantification of processed SREBP1 levels were presented as an average of three independent samples. The protein levels were normalized to tubulin and were shown in fold change from LFD WT. **d** Protein lysates from (**b**) were analyzed for pAKT, total AKT, pPKCβII, pPKCζ, pPKCε, and tubulin using western blotting. **e, f** Quantification of pAKT and pPKCβII protein from (**d**) ($N = 3$). **g–i** The *Inpp4b*$^{-/-}$ animals were treated with vehicle DMSO ($N = 4$), 6 mg/kg of AZD5363 ($N = 2$) or BIM-I ($N = 4$) for 16 h. Livers were dissected and used for protein and RNA purification. One-way ANOVA (Dunnett's multiple comparison test) was used. (**g**) Fifty μg of protein extracted from livers of *Inpp4b*$^{-/-}$ animals treated with vehicle, BIM-I, or AZD5363 were used to compare levels of unprocessed and processed forms of SREBP1 and tubulin. (**h**) Quantification of processed or unprocessed SREBP1 proteins in (**g**) ($N = 4$). **i** RNAs purified from livers of *Inpp4b*$^{-/-}$ animals treated with vehicle, BIM-I, or AZD5363 were used to compare expression of SREBP1 target genes *Mogat1*, *Cyp2b13*, and *Cyp2b9* by qRT-PCR using *18S* as control. **j** WT or *Inpp4b*$^{-/-}$ mice were fasted overnight. Protein lysates from fasted and fed mice were isolated and analyzed for pAKT, precursor and processed forms of SREBP1, and tubulin using western blotting. **k, l** Quantification of pAKT and processed SREBP1 protein levels from two independent experiments ($N = 4$ for each individual bar) ($^*p < 0.05$, $^{**}p < 0.01$, $^{***}p < 0.001$, $^{****}p < 0.0001$. Exact $p$ values are provided in the Source data).

fibromuscular membranes (Fig. 7a, b and Supplementary Fig. 6b). Distribution of E-cadherin in normal prostate glands was restricted to the lateral and basal membranes of the secretory epithelium. In neoplastic glands of HFD *Inpp4b*$^{-/-}$, epithelial cells lost their polarity and E-cadherin distributed evenly around the cellular membranes (Supplementary Fig. 6a). Similar to previously reported *c-myc*-induced neoplastic lesions in the mouse prostate[76,77], neoplastic lesions in HFD *Inpp4b*$^{-/-}$ males display hypochromatic, enlarged atypical nuclei with prominent enlarged nucleoli when compared to the more compact and densely stained nuclei of normal prostate epithelium cells (Supplementary Fig. 6c).

To the best of our knowledge, we discovered a novel function of INPP4B. We showed that INPP4B is an essential regulator of metabolic health. INPP4B expression protects the liver from steatosis, mediates insulin sensitivity, and links obesity to neoplastic changes in the prostate epithelium. As shown in Fig. 8, *Inpp4b*-deficient animals developed hyperglycemia on normal chow and T2D on HFD due to decreased insulin sensitivity. The loss of INPP4B induced the activation of hepatic SREBP1, ultimately leading to the development of NAFLD. Only the loss of INPP4B in HFD-fed mice promoted prostate neoplasia, suggesting that INPP4B might provide a functional link between obesity and increased prostate cancer incidence. In both mice and men, a decline in hepatic INPP4B expression correlated with increasing severity of NAFLD. Thus, *Inpp4b*$^{-/-}$ males faithfully reproduce the complex metabolic syndrome in men and, accordingly, the INPP4B signaling cascade should be considered for therapeutic intervention in this wide-spread disease.

## Methods

**Animal studies.** Animals were housed at AAALAC certified facility at Florida International University. All protocols were approved by Florida International University Institutional Animal Care and Use Committee. *Inpp4b*$^{-/-}$ mice were provided by Dr. Vacher and bred into an FVB background[56,78]. WT and *Inpp4b*$^{-/-}$ males were fed with either LFD (LabDiet 5V75, St. Louis, MO) or HFD (TestDiet 58R3, St. Louis, MO). The HFD consisted of 59.4% fat, 25.7% carbohydrate, and 14.9% protein (total 22.8 kJ/g), whereas the regular chow contained 12.9% fat, 63.8% carbohydrate, and 23.2% protein (total 13.6 kJ/g). LFD groups were continuously maintained on LFD. For HFD cohorts, dams were fed the designated diet for 1 month prior to mating, during pregnancy, and 3 weeks lactation. Male pups were maintained on the HFD until euthanasia[79,80]. Mouse body weights were measured once a week. All LFD WT, LFD *Inpp4b*$^{-/-}$, HFD WT, and HFD *Inpp4b*$^{-/-}$ mice were euthanized and dissected at 12 weeks and their tissues were collected for analysis. Six-month-old *Inpp4b*$^{-/-}$ mice were treated with DMSO control, or 6 mg/kg of Akt inhibitor AZD5363 (SelleckChem, Houston, TX) or PKC inhibitor BIM-I (AdipoGen, San Diego, CA) by intraperitoneal injection. After 16-h treatment, mice were euthanized, and livers were collected for gene and protein analyses.

**Hemodynamic measurements.** Eleven-week-old HFD, WT, and *Inpp4b*$^{-/-}$ mice were anesthetized with 2% isoflurane in 100% oxygen. Each mouse was placed on a heating platform with the temperature between 38 and 40 °C. A tail cuff connected to the pressurizing tubing was placed at the base of the mouse tail and attached to the CODA monitor (Kent Scientific, Torrington, CT). The heart rate, systolic, diastolic, and mean blood pressure where recorded using a non-invasive blood pressure system Coda Monitor Noninvasive Blood Pressure System with Coda Mouse Cuff Kit (Kent Scientific, Torrington, CT).

**Indirect calorimetry and body composition analysis.** Indirect calorimetry and body composition analysis were performed at Vanderbilt Mouse Metabolic Phenotyping Center. Eight WT and ten *Inpp4b*$^{-/-}$ males were individually placed in metabolic cages (identical to home cages with bedding) in a 12 h light/dark cycle, temperature/humidity-controlled dedicated room located in the Vanderbilt MMPC. Energy expenditure measures were obtained by indirect calorimetry (Promethion, Sable Systems, Las Vegas, NV). The calorimetry system consists of cages identical to home cages with bedding equipped with water bottles and food hoppers connected to load cells for food and water intake monitoring. All animals had ad libitum access to standard rodent chow and water. The air within the cages is sampled through microperforated stainless-steel sampling tubes that ensure uniform cage air sampling. Promethion utilizes a pull-mode, negative pressure system with an excurrent flow rate set at 2000 ml/min. Water vapor is continuously measured and its dilution effect on $O_2$ and $CO_2$ are mathematically compensated for in the analysis stream[81]. $O_2$ consumption and $CO_2$ production are measured for each mouse every 5 min for 30 s. Incurrent air reference values are determined every 4 cages. Respiratory quotient (RQ) is calculated as the ratio of $CO_2$ production over $O_2$ consumption. Energy expenditure is calculated using the Weir equation: EE (kcal/h) = $60 \times (0.003941 \times \text{VO}_2(\text{ml/min}) + 0.001106 \times \text{VCO}_2(\text{ml/min}))$[82]. Ambulatory activity was determined every second with XYZ beams. Data acquisition and instrument control were coordinated by MetaScreen v2.2.18 and the raw data were processed using ExpeData v1.7.30 (Sable Systems). Body composition in awake animals was determined at the Vanderbilt Mouse Metabolic Phenotyping Center by NMR using Bruker Minispec body composition analyzer (Bruker Optics, Billerica, MA).

**Gene expression analysis.** RNA was isolated from tissues using Tri Reagent (Molecular Research Center, Cincinnati, OH) and reverse transcribed using a Verso cDNA synthesis Kit (Thermo Fisher Scientific, Waltham, MA). The quantitative real-time PCR was performed using primers and probes from Universal Probe Library (Supplementary Data 1) (Roche, Basel, Switzerland) and a Roche 480 LightCycler (Roche). For gene expression analysis, samples were from LFD WT ($N = 12$), LFD *Inpp4b*$^{-/-}$ ($N = 11$), HFD WT ($N = 9$), and HFD *Inpp4b*$^{-/-}$ ($N = 11$) mice.

**Glucose tolerance test (GTT) and pyruvate tolerance test (PTT).** The glucose tolerance tests were performed on 11-week-old males as previously described[83]. Briefly, after 6 h of fasting, mice ($N \geq 8$) were weighed and given 2 g/kg of glucose (Sigma-Aldrich, St. Louis, MO) through a gavage needle (18 G, 5.08 cm; Cadence Science, Cranston, RI). A drop of blood was sampled from each mouse at 0, 15, 30, 60, 90, and 120 min after oral gavage and the glucose concentration was determined immediately using the ACCU-CHEK Nano SmartView glucometer (Roche Diagnostics, Indianapolis, IN). For pyruvate tolerance testing, mice were fasted overnight, followed by intraperitoneal injection of pyruvate (1.5 g/kg, Sigma-Aldrich, St. Louis, MO). The blood glucose levels were measured as above.

**Insulin tolerance test (ITT).** Four-month-old LFD WT ($N = 5$) and LFD *Inpp4b*$^{-/-}$ ($N = 9$) male mice were fasted for 3 h in clean cages with standard light/dark cycle and free access to water. The mice were weighed and given 0.75 U/kg Humulin R insulin (Eli Lilly and Company, Indianapolis, IN) by intraperitoneal injection. A drop of blood was sampled from each mouse immediately before insulin injection and 30, 60, 90, and 120 min after. Glucose concentrations were determined using the ACCU-CHEK Nano SmartView glucometer (Roche Diagnostics, Indianapolis, IN).

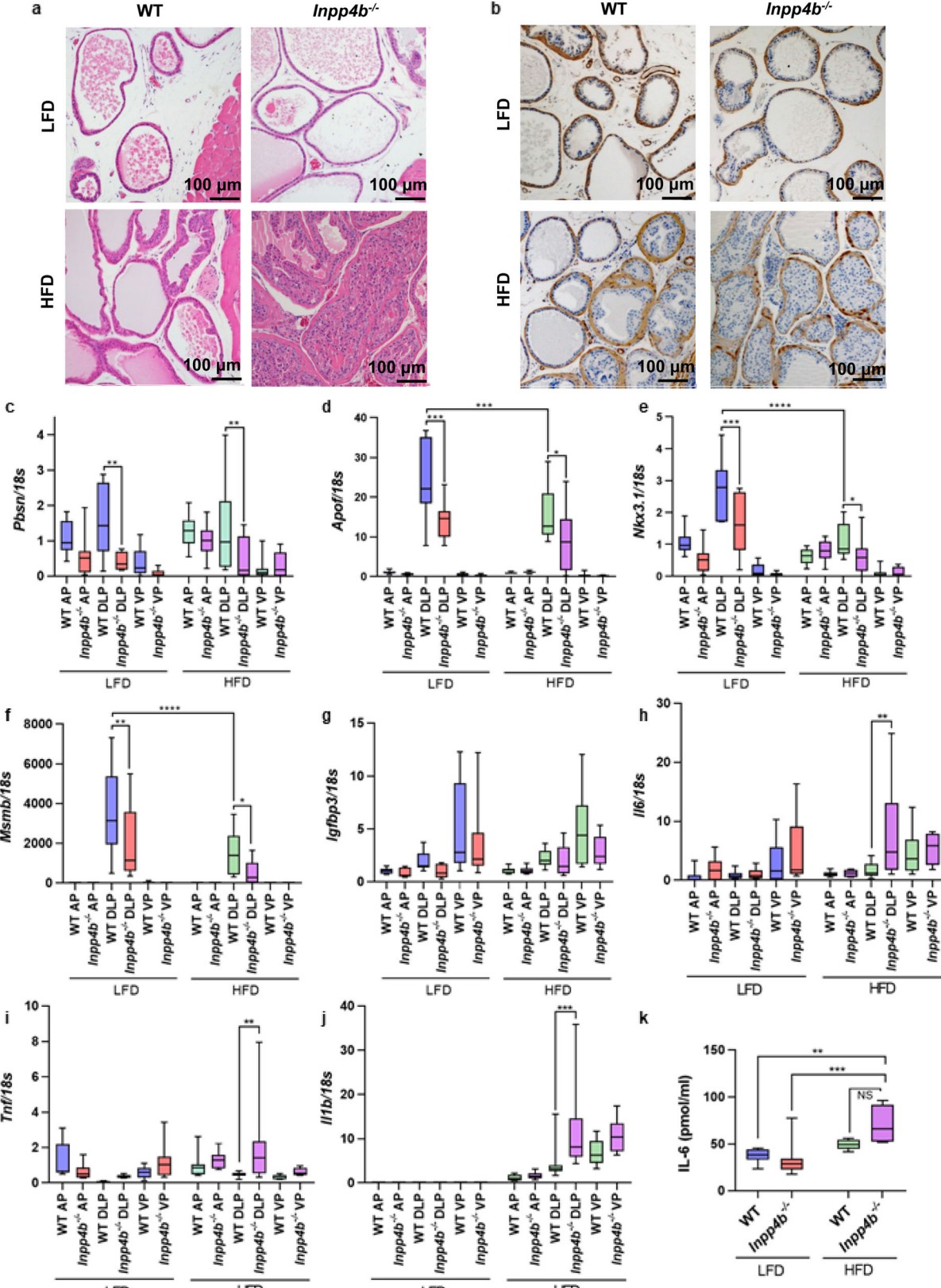

**Fig. 7 Loss of *Inpp4b* leads to the development of PIN and inflammation in obese males. a** H&E staining of prostates from LFD WT, LFD *Inpp4b*−/−, HFD WT, and HFD *Inpp4b*−/− males. **b** Representative tissue sections from LFD WT, LFD *Inpp4b*−/−, HFD WT, and HFD *Inpp4b*−/− prostates were stained for α-SMA and counterstained with hematoxylin. Anterior prostate (AP), dorsolateral prostate (DLP), and ventral prostate (VP) were dissected from LFD WT ($N = 8$), LFD *Inpp4b*−/− ($N = 8$), HFD WT ($N = 9$), and HFD *Inpp4b*−/− ($N = 9$) mice. **c–j** RNA was extracted and the expression levels of *Pbsn* (**c**), *Apof* (**d**), *Nkx3.1* (**e**), *Msmb* (**f**), *Igfbp3* (**g**), *Il6* (**h**), *Tnf* (**i**), and *Il1b* (**j**) were compared by qRT-PCR using *18S* as an internal control. **k** Protein concentrations of IL-6 in mouse DLPs were measured using IL-6 ELISA ($^*p < 0.05$, $^{**}p < 0.01$, $^{***}p < 0.001$, $^{****}p < 0.0001$. Exact $p$ values are provided in the Source data).

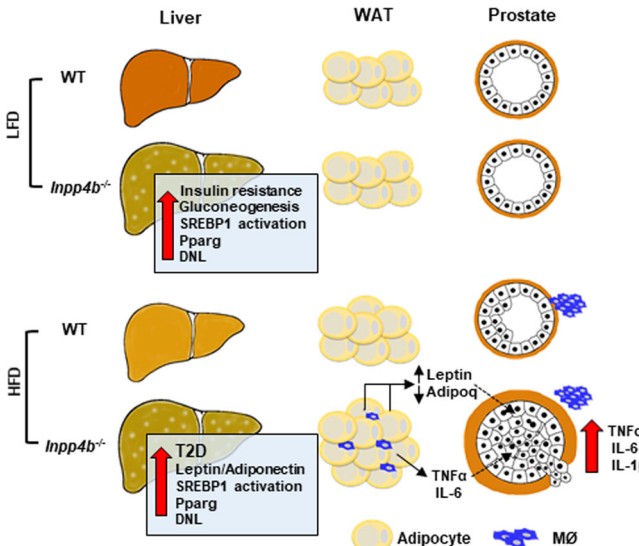

**Fig. 8 Physiological functions of *INPP4B* in liver, WAT, and prostate.** The loss of INPP4B activates SREBP1, stimulates hepatic expression of *Pparg*, causing liver steatosis, insulin resistance, de novo lipogenesis (DNL), and an increased rate of gluconeogenesis. When consuming a HFD, the *Inpp4b*−/− males developed type 2 diabetes, inflammation, and macrophage (MØ) infiltration of WAT, DNL, and an increased leptin/adiponectin ratio, all of which were shown to stimulate the development of PIN. Consumption of a HFD promoted macrophage infiltration in the prostates of both WT and *Inpp4b*−/− mice; however, increased production of proinflammatory cytokines was only evident in dorsolateral lobes of prostates of *Inpp4b*−/− mice.

**ELISA and adipokine array.** Blood insulin levels were determined using the Ultra Sensitive Mouse Insulin ELISA Kit (Crystal Chem, Elk Grove Village, IL) according to the manufacturer's instructions. Briefly, 5 µl of blood serum from LFD WT ($N = 23$), LFD *Inpp4b*−/− ($N = 20$), HFD WT ($N = 9$), and HFD *Inpp4b*−/− ($N = 9$) mice were used for the test. The insulin concentrations were measured using a CLARIOstar plate reader (BMG Labtech, Cary, NC). A Mouse IL6 ELISA kit (Sigma-Aldrich) was used to determine the IL6 level in LFD WT ($N = 6$), LFD *Inpp4b*−/− ($N = 10$), HFD WT ($N = 5$), and HFD *Inpp4b*−/− ($N = 5$) mouse DLPs. Adipokines and proinflammatory cytokines were detected in WAT protein extracts using the Proteome Profiler Mouse Adipokine Array Kit (R&D Systems, Minneapolis, MN). Samples from five mice in each group were pooled and used for analysis. The signal was captured using an ImageQuant LAS 500 imager (GE Healthcare, Chicago, IL).

**Multiplex analysis of metabolic cytokines and hormones.** The serum from LFD WT ($N = 11$), LFD *Inpp4b*−/− ($N = 11$), HFD WT ($N = 9$), HFD *Inpp4b*−/− ($N = 9$) male mice was analyzed for the following cytokines and chemokines: Amylin (active), C-Peptide 2, GIP (total), GLP-1(active), Ghrelin (active), Glucagon, Insulin, Leptin, PP, PYY, and Resistin. The analysis was done using the Mouse Metabolic 11-plex Arrays (Eve Technologies, Calgary, AB, Canada).

**Morphometric quantification of steatosis.** Liver steatosis was evaluated by measuring the surface of the unstained area of the H&E stained liver sections per field using 40X magnification. Five mice were selected from each group and 10 different areas from each mouse were counted using ImageJ software (National Institute of Mental Health, Bethesda, USA).

**Oil red O staining.** Fresh liver tissue was placed in OCT medium and snap frozen in liquid nitrogen. Liver samples were cut into 12 µm sections and attached to glass slides. Oil red O was dissolved in hot propylene glycol and filtered through Whatman #2 filter paper. Slides were fixed in 4% PFA, rinsed with distilled water and propylene glycol followed by staining with Oil red O and counterstained with hematoxylin. All images were acquired using an AxioCam MRc5 camera (Zeiss, Thornwood, NY).

**Determination of triglycerides in mouse liver.** The total triglycerides in liver were measured in LFD WT ($N = 8$), LFD *Inpp4b*−/− ($N = 8$), HFD WT ($N = 6$), and HFD *Inpp4b*−/− ($N = 4$) mice using a Triglyceride Colorimetric Assay Kit (Cayman Chemical, Ann Arbor, MI). The assay was performed according to the manufacturer's instruction. The total amount of triglycerides in liver was calculated.

**RNA sequencing and data analysis.** RNA was isolated as described above from HFD WT ($N = 4$) and HFD *Inpp4b*−/− ($N = 4$), and submitted to Novogene for RNA sequencing. The libraries were generated using the NEBNext® Ultra™ RNA Library Prep Kit and used for Illumina 150-bp paired-end sequencing. Quality control assessment was done using Illumina RNA-seq pipeline to estimate genomic coverage, percent alignment, and nucleotide quality. Raw reads were mapped to the reference mouse genome (the most recent build GRCm38) using HISAT2[84] and STAR[85] software. The reads for each gene aligned by HISAT2 were counted using HTSeq software[86]. Alignment by STAR was run with the option "—quantMode Gene Counts" that allowed counting of reads aligned to each gene. Raw counts from HTSeq and STAR were imported into Bioconductor/R package DESeq2[87], normalized, and tested for differential gene expression. This analysis was done separately for the files produced by each aligner. In each analysis we selected genes that were differentially expressed based on the criteria of false discovery rate (FDR) <10% and fold change >1.3 in either direction. Genes that showed differential expression after analysis of the files from both aligners have been selected for further analysis. Data has been uploaded to NCBI's gene expression omnibus (GEO) database with accession number GSE134466.

**Pathway analysis and Gene Ontology (GO) analysis.** Biological processes and pathways affected by INPP4B knockout in mouse liver were performed using GSEA (http://www.broad.mit.edu/gsea/index.html) and DAVID (the Database for Annotation, Visualization and Integrated Discovery) analysis with DEGs obtained from the RNA-seq analysis. The gene sets for KEGG pathways (c2 KEGG curated) and GO analysis (c5 curated) were acquired from Molecular Signatures Database (http://software.broadinstitute.org/gsea/msigdb/index.jsp). The normalized enrichment score (NES) with FDR < 25% or p value < 0.05 was considered significant[88].

**Immunohistochemistry.** Paraffin-embedded tissues were sectioned at 5 µM, and deparaffinized. For H&E, sections were then stained with Harris' Alum Hematoxylin (#638A-85, Millipore, Burlington, MA) and Eosin-Y-Solution (#314−630, Thermo Fisher Scientific)[56]. For immunohistochemistry, antigen retrieval was done by heating slides at 99 °C for 15 min in 10 mM citrate buffer, pH = 6. Sections were blocked in 1% $H_2O_2$ in water and 5% BSA in PBS. Primary antibodies were diluted in PBS at 1:6000 for α-SMA (#ab5694, AbCam, Cambridge, MA), 1:60,000 for insulin (#ab181547, AbCam), 1:500 for E-cadherin (#3195, Cell Signaling, Danvers, MA), and 1:200 in Ki67 (#RB-1510-P1, Thermo Fisher Scientific). Antibodies were incubated overnight at 4 °C or for 1 h at room temperature. Sections were washed and incubated with biotinylated anti-rabbit secondary antibody for 18 min, followed by incubation with streptavidin-conjugated peroxidase for 30 min using the Vectastain ABC Kit (Vector Laboratories, Burlingame, CA). The staining was developed using the ImmPACT DAB Peroxidase Substrate Kit (Vector Labs, Burlingame, CA) and then the slides were counterstained with hematoxylin.

**Western blotting.** Mouse tissues were homogenized with a glass grinder in ice-cold RIPA buffer with protease and phosphatase inhibitors (GenDepot, Barker, TX). Cleared lysates were diluted to 4 µg/µl and 15–30 µg of protein was resolved on a 7.5–10% SDS-PAGE. For immunoblotting, the following primary antibodies and dilutions were used: rabbit antibodies total Akt (1:1000, #4691, Cell Signaling, Danvers, MA), pPKCζ T410 (1:1000, # 2060, Cell Signaling), pPKCβII S660 (1:1000, #9371, Cell Signaling), FAS (1:1000, #3180, Cell Signaling), HK2 (1:1000, #2867, Cell Signaling), PTEN (1:1000, #9188, Cell Signaling), primary mouse antibodies pAkt S473 (1:1000, #4051, Cell Signaling), β-tubulin (1:5000, #05−661, Millipore), SREBP1 (1:1000, NB #600-582, Novus Biologicals, Littleton, CO), CD68 (1:800, #ab 125212, AbCam, Cambridge, MA), and primary sheep antibody PKCε (1:2000, AF5134, R&D Systems, Minneapolis, MN). The following secondary antibodies were used: rabbit IgG (1:2000, W4011, Promega, Madison, WI), mouse IgG (1:2000, W4021, Promega), and sheep IgG (1:1000, HAF016, R&D systems). The signal was captured using an ImageQuant LAS 500 imager and analyzed by ImageQuant TL software (GE Healthcare, Chicago, IL).

**Statistics and reproducibility.** All data are presented as mean ± SEM, unless otherwise stated. Two-way ANOVA (Sidak multiple comparison test) for 4 groups and two-tailed Student's t-test for 2 groups were performed by Prism 7.0. The p value < 0.05 and confidence level of 95% were considered statistically significant. Individual p values are reported in figure legends and/or in Source data (reported in Supplementary Data 2). For the box plots, the whiskers go down to the smallest value and up to the largest. The box extends from the 25th to 75th percentiles. Center line is plotted at the median.

**Reporting summary**. Further information on research design is available in the Nature Research Reporting Summary linked to this article.

## Data availability
Source data are available in Supplementary Data 2. RNA-sequencing data have been deposited in NCBI's Sequence Read Archive (SRA) database with accession number GSE134466. The datasets that support the findings of this study are available in (https://www.ncbi.nlm.nih.gov/geo/) with the accession numbers and DOIs: GSE15653 (https://doi.org/10.1210/jc.2009-0212), GSE126848 (https://doi.org/10.1152/ajpgi.00358.2018), GSE89091 (https://doi.org/10.1038/s41374-018-0088-6) and GSE24335 (https://doi.org/10.1101/gr.112821.110). Source data for Figs. 1–7 and Supplementary Figs. 1, 3, 4, 5, 7, 8 are available in Supplementary Data 2.

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

## Acknowledgements

This work was supported by NCI grant R15 CA179287-01A1 (I.U.A. and M.Z.), NIDDK 1R01DK110167 (A.I.A., I.U.A., and E.M.K.), and Canadian Institutes of Health Research grant #123343 (J.V.). The indirect calorimetry study was performed by the Vanderbilt Mouse Metabolic Phenotyping Center (DK059637 and 1S10RR028101-01).

## Author contributions

I.U.A. designed and supervised the study; A.I.A. provided expertise for the mouse hepatosteatosis studies; M.M.I. conducted comparative histological analysis of the WT and the *Inpp4b*$^{-/-}$ mouse prostates; M.Z., Y.C., E.M.K., and J.L.V. performed experiments and prepared the figures; L.N. has conducted RNA-seq, bioinformatics, and data analysis; J.V. provided the *Inpp4b*$^{-/-}$ mouse model; F.K.K. provided expertise in transcriptomics in normal and steatotic human livers; I.U.A. and M.Z. wrote the manuscript with substantive contributions from all co-authors.

## Competing interests

The authors declare no competing interests.
