## [Peer Review File · Communications Biology]

Reviewers' comments:

Reviewer #1 (Remarks to the Author):

In this manuscript the authors have identified a metabolic phenotype of INPP4B knockout mice, which has not been previously reported. The authors show that INPP4B knockout mice fed a high fed diet developed metabolic dysfunction characterized by hyperglycemia, liver steatosis and weight gain. They also show that INPP4B knockout mice fed a high fat diet developed systemic inflammation leading to early onset prostate intraepithelial neoplasia. These findings uncover a novel role for INPP4B in maintaining metabolic homeostasis in vivo. However, further experiments are required to characterize the metabolic phenotype of INPP4B knockout mice, and to demonstrate the signaling pathways that mediate this phenotype.

Major issues

1. Further metabolic profiling experiments need to be conducted to characterise the metabolic dysfunction in LFD and HFD INPP4B KO versus WT mice. Food intake and energy expenditure should be examined, and an insulin tolerance test should be performed to determine whether INPP4B KO mice are insulin resistant. In addition, in vitro adipocyte differentiation assays should be performed to validate the effects of INPP4B KO on adipogenesis. Based on the data provided in the manuscript it would be more accurate to describe the phenotype of HFD INPP4B KO as diabetic-like, rather than as type 2 diabetes. Although there is evidence of hyperglycemia, the authors do not demonstrate insulin resistance (eg insulin tolerance test) or pancreatic dysfunction.
2. The authors claim that “the activation of insulin, retinol, AKT and PKC pathways in obese mice knockout males stimulates proteolytic cleavage of SREBP1 in the livers of HFD Inpp4b^{-/-} group activating expression of Pparg and other lipogenic enzymes and leading to the development of hepatosteatosis”. However, the authors do not demonstrate any cause and effect between AKT/PKC signaling, SREBP1 processing and downstream lipogenic gene expression following loss of INPP4B expression. Use of AKT inhibitors and PKC inhibitors in primary liver cells or immortalised liver cell lines (eg HepG2, THLE-2 or AML12 cells) should be used to determine whether the effects of INPP4B KO on SREBP1 cleavage and lipogenic gene expression can be restored by inhibition of these pathways.
3. Although the authors have shown transcriptional changes in inflammatory cytokines and leptin/adiponectin in INPP4B KO mice, they have not shown that secretion of the corresponding proteins is affected by INPP4B KO. This should be performed by measuring TNF α , IL6 and leptin/adiponectin protein levels in the serum of mice.
4. The conclusions drawn from the prostate pathology, specifically “loss of polarity, nuclear atypia and a discontinuous fibromuscular layer around individual glands”(line 219), are not supported by the data provided (Figure 6a and 6b). Higher power images are required to visualize nuclear atypia. Loss of polarity needs to be confirmed with polarity marker staining (eg E-cadherin). The SMA staining images do not appear to show any discontinuous staining as is described in the text. In addition, the presence of several liver histopathological features, specifically microvesicular steatosis and Mallory bodies, also cannot be verified from the low resolution images provided (Figure 3e).

Minor issues

1. The authors state “INPP4B suppresses AKT and PKC signaling in liver by dephosphorylating PI(3,4)P2 and PI(4,5)P2 respectively thereby improving insulin sensitivity” (line 28). If the authors claim that the phenotype is dependent on PI(4,5)P2 then they need to show increased PI(4,5)P2 levels in INPP4B KO cells, as no studies have previously reported that INPP4B regulates cellular PI(4,5)P2 levels.
2. Several references are missing in introduction (line 41, line 43).
3. INPP4B deletion should be validated by qPCR, western blot or IHC staining.
4. In Figure 1B and Supplementary table 1, it is not indicated whether there is a statistical difference in body weight between HFD fed WT and INPP4B-KO mice as described in the text (line 89).
5. Leptin and adiponectin (line 106) should be in italics as this statement is referring to the genes and not the proteins.
6. The authors state “The array analysis confirmed the increased leptin/adiponectin ratio at the protein level” (line 115). The mRNA data, which showed that only INPP4B-KO HFD mice had an increased lep/adipo ratio, does not match with the array analysis data, which showed that INPP4B-KO LFD, WT HFD and INPP4B-KO HFD mice all had an increased lep/adipo ratio to a similar level, and this should be acknowledged.
7. The glucose and pyruvate tolerances tests in Figure 2a and Supplementary Figure 2a should also be quantified as area under the curve, and statistical comparisons between all groups should be performed.
8. Figure 5d-f are not described in the text.
9. The Ki67 staining in Supplementary Figure 5a does not appear to be altered in INPP4B WT versus KO HFD mice as the authors state (line 226), and quantification is needed to support this conclusion.
10. The authors state “Our data suggest that fasting-induced shut off of SREBP1 (typo) processing depends on functionally active INPP4B” (line 272). If the authors claim that INPP4B functional activity is required for SREBP1 processing then they need to perform rescue experiments with expression of wild-type and catalytically inactive INPP4B.
11. The authors state “Similar to our HFD Inpp4b^{-/-} mouse model, IL6 is elevated in the serum of patients with metastatic PCa” (line 285). As serum IL6 levels were not measured in mice, this experiment should be performed to confirm increased IL6 secretion.
12. Several quantifications do not match the relevant western blots (Figure 5g and 5i, Supplementary Figure 4a and 4b).
13. For western blots where multiple bands are present (Figure 5d and 5g), the band(s) of interest should be labeled.
14. Several graphs are missing labels or units (Figures 4a and 4b, Supplementary Figures 1b and 1c).

15. The depicted statistical comparisons in Supplementary Figure 5d and Figure 6k are not between the groups that the authors describe in the text. Statistical comparisons between WT and INPP4B-KO HFD mice should be depicted.

Reviewer #2 (Remarks to the Author):

In the manuscript entitled "INPP4B protects from metabolic disease and associated disorders" by Zhang and colleagues, the authors investigate the consequences of total ablation of the phosphatase INPP4B on metabolic disease and on prostatic intraepithelial neoplasia. While the basic phenotypes described, including increased body weight, fatty liver, altered GTT and presence of PIN in mice with deletion of this phosphatase are potentially interesting, the studies reported to characterize them physiologically and mechanistically appear still preliminary at this stage and require further in depth characterization.

1) The characterization of the fat phenotype of the KO in Fig 1 appears incomplete.

a) The authors should report the levels of INPP4B in all fat depots, not only of eWAT, to provide a clearer picture of the possible function of INPP4B in those depots.

b) The obesity phenotype in KO mice eluded to by the authors does not appear sufficiently supported by the differences shown in total body weight. Authors should provide information on fat and lean mass of WT and KO.

c) Weights of all fat depots, not just of eWAT of KO mice compared to WT, should be provided.

d) Complete pathological and morphological characterization of all fat depots by H and E staining should be presented.

e) Data on adipose cell size of WT and KO fat tissues to assess whether hyperplasia and/or hypertrophy is present in the KO should be added.

2) The authors claim that KO mice have increased inflammation in fat based on levels of some markers. Analysis of macrophage infiltration into fat tissues of WT and KO should be provided as a further evidence.

3) Graphs of the expression of different genes in the same tissue should be grouped in the same panel to make the results section less scattered and unfocused.

4) ITT should be performed and the effects of ablation of INPP4B on classic insulin signaling pathways should be described to definitively conclude impaired insulin sensitivity.

5) The data provided on WT and KO livers are scarce. Further measurements to characterize the liver phenotype, including measurements of TG and quantification over multiple sections of liver biopsies of WT and KO of Mallory bodies (and other morphological changes reported) and of markers of inflammation (i. e: F4/80) should be provided.

6) The data on PIN is very preliminary. More detailed characterization of PIN should be provided.

7) The mechanistic explanation of the protective role of INPP4B in metabolism is tentative. The link between SREBP -PPAR γ the authors invoke is not clear.

We are grateful for Reviewers' careful consideration and thoughtful comments. We believe that addressing them significantly strengthened our manuscript.

Reviewers' comments:

Reviewer #1 (Remarks to the Author):

In this manuscript the authors have identified a metabolic phenotype of INPP4B knockout mice, which has not been previously reported. The authors show that INPP4B knockout mice fed a high fed diet developed metabolic dysfunction characterized by hyperglycemia, liver steatosis and weight gain. They also show that INPP4B knockout mice fed a high fat diet developed systemic inflammation leading to early onset prostate intraepithelial neoplasia. These findings uncover a novel role for INPP4B in maintaining metabolic homeostasis in vivo. However, further experiments are required to characterize the metabolic phenotype of INPP4B knockout mice, and to demonstrate the signaling pathways that mediate this phenotype.

We are grateful to the Reviewer #1 for acknowledgment the novelty of our findings.

Major issues

Q1: Further metabolic profiling experiments need to be conducted to characterise the metabolic dysfunction in LFD and HFD INPP4B KO versus WT mice. Food intake and energy expenditure should be examined, and an insulin tolerance test should be performed to determine whether INPP4B KO mice are insulin resistant.

A: As requested by the Reviewer #1, we sent WT and *Inpp4b*^{-/-} males to Vanderbilt Mouse Metabolic Phenotyping Center to determine food and liquid intake, body composition, ambulatory activity, energy expenditure, and respiratory exchange ratios. The results showed that *Inpp4b*^{-/-} males have increased fat to lean body mass ratio. Even though, knockout males consumed slightly less food and water, decreased ambulatory activity, energy expenditure, and respiratory exchange ratio resulted in weight gain, fat mass in particular was increased in *Inpp4b*^{-/-} males (Figures 1d – k, and supplemental figures 1f – g). We also performed insulin tolerance test showing that *Inpp4b*^{-/-} males are insulin resistant (Supplementary figure 4a).

Unfortunately, we were unable to include HFD males in this evaluation, because creating this group requires maintaining dams on high fat diet prior to breeding, then breeding to obtain experimental animals, and ageing them. Due to the length of the breeding protocol and restrictions imposed by quarantine closeouts of the university and animal facility, we were unable to produce HFD groups. However, the data in LFD showed striking differences between mutant and wild-type mice providing mechanistic insight into described phenotype.

Q2: In addition, in vitro adipocyte differentiation assays should be performed to validate the effects of INPP4B KO on adipogenesis.

A: We examined histology of the Inguinal, retroperitoneal, mesenteric, and epididymal WAT and found no significant changes in adipose cell morphology. We also examined expression of key adipocyte differentiation markers *Pparg*, *Srebf*, and *Fasn*. PPAR γ in particular is considered a master regulator of adipogenesis, without it, precursor cells are incapable of expressing any

known aspect of the adipocyte phenotype^{1,2}. No difference was observed in *Pparg*, and *Fasn* expression between WT and *Inpp4b*^{-/-} animals (see below). There was small but significant difference in *Srebf1* expression in WAT of HFD WT and HFD *Inpp4b*^{-/-} males which was much less than the difference in expression in previously reported set GSE20752³. These data suggest high degree of similarity in adipocyte differentiation between WT and knockout animals

Q3: Based on the data provided in the manuscript it would be more accurate to describe the phenotype of HFD INPP4B KO as diabetic-like, rather than as type 2 diabetes. Although there is evidence of hyperglycemia, the authors do not demonstrate insulin resistance (eg insulin tolerance test) or pancreatic dysfunction.

A: As requested by the Reviewer we conducted insulin tolerance test (Supplemental figure 4a) and found knockout males to be resistant to insulin. Based on our observations that *Inpp4b*^{-/-} males are insulin resistant, severely hyperglycemic, and have elevated insulin levels. Based on classification proposed by Clee and Attie⁴ stating that the fasting glucose level exceeding 250 mg/dL is the threshold for diabetes mellitus in mice, we concluded that HFD *Inpp4b*^{-/-} males fit diabetes mellitus profile (Figures 3a and 3d, Supplementary figure 3a).

Q4: The authors claim that “the activation of insulin, retinol, AKT and PKC pathways in obese mice knockout males stimulates proteolytic cleavage of SREBP1 in the livers of HFD *Inpp4b*^{-/-} group activating expression of *Pparg* and other lipogenic enzymes and leading to the development of hepatosteatosis”. However, the authors do not demonstrate any cause and effect between AKT/PKC signaling, SREBP1 processing and downstream lipogenic gene expression following loss of INPP4B expression. Use of AKT inhibitors and PKC inhibitors in primary liver cells or immortalised liver cell lines (eg HepG2, THLE-2 or

AML12 cells) should be used to determine whether the effects of INPP4B KO on SREBP1 cleavage and lipogenic gene expression can be restored by inhibition of these pathways.

A: There is a substantial body of literature that shows Akt and PKC regulation of SREBP1 expression and processing⁵⁻¹¹. As suggested by the reviewer, we attempted treatments of the mouse primary liver cells with Akt and PKC inhibitors to determine Akt and PKC pathways' role in SREBP1 expression and processing in our model. However, similar to previous report¹², primary hepatocytes rapidly lost expression of SREBP1 in culture. In the figure below (left), primary hepatocytes were plated on the tissue culture plates with or without gelatin coating and after 24 and 48 hours of incubation, SREBP1 protein was nearly undetectable. Similar level of SREBP1 degradation was observed after 30 min incubation of primary hepatocytes with or without insulin treatment (below right).

Therefore, we treated live mice with Akt and PKC specific inhibitors, AZD5363 and BIM-1, for 16 hours and found that levels of both processed and unprocessed SREBP1 significantly declined and the lipogenic gene expression regulated by SREBP1 was also significantly decreased (Figures 6g – i). We have added the statement that Akt and PKC are required for maintenance of unprocessed and mature forms of SREBP1 proteins and its transcriptional activity.

Q5: Although the authors have shown transcriptional changes in inflammatory cytokines and leptin/adiponectin in INPP4B KO mice, they have not shown that secretion of the corresponding proteins is affected by INPP4B KO. This should be performed by measuring TNF α , IL6 and leptin/adiponectin protein levels in the serum of mice.

A: We have analyzed serum from LFD WT (N = 11), LFD *Inpp4b*^{-/-} (N = 11), HFD WT (N = 9), HFD *Inpp4b*^{-/-} (N = 9) male mice using Mouse Metabolic Hormone Multiplex Discovery Assay and Mouse Multiplex Inflammatory Cytokine Assay from Eve Technologies (Calgary, AB, Canada). We found significant changes in four metabolic hormones and these data are now presented in Figures 2i – l and Supplemental figure 2e. We however did not see statistically significant changes in inflammatory cytokines, IL-6, TNF α , IL-1 β , in mouse serum which we noted in the discussion section.

Q6: The conclusions drawn from the prostate pathology, specifically “loss of polarity, nuclear atypia and a discontinuous fibromuscular layer around individual glands”(line 219), are not supported by the data provided (Figure 6a and 6b). Higher power images are required to visualize nuclear atypia. Loss of polarity needs to be confirmed with polarity marker staining (eg E-cadherin). The SMA staining images do not appear to show any discontinuous staining as is described in the text. In addition, the presence of several liver histopathological features, specifically microvesicular steatosis and Mallory bodies, also cannot be verified from the low resolution images provided (Figure 3e).

A: As suggested by the reviewer we stained mouse prostates with E-cadherin antibody to evaluate the loss of polarity in prostatic lesions. E-cadherin is a key component of adherens junctions and therefore is found only in basal and lateral membranes of prostate secretory epithelium in a normal prostate¹³. In neoplastic lesions of HFD *Inpp4b*^{-/-} males the distribution was detected throughout the whole membrane (Supplemental figure 6a) suggesting the loss of polarity of luminal epithelial cells.

We have taken high power images of the nuclei within hyperplastic epithelial regions (Supplemental figure 6c). Similar to previously published reports on murine prostate carcinomas caused by C-MYC^{14,15} overexpression, neoplastic lesions featured enlarged hypochromatic nuclei with prominent nucleoli distinctly different from smaller densely stained nuclei of the normal prostate epithelium.

We have also showed enlarged images of two glands where epithelial cells have escaped prostatic gland into the stromal compartment (Supplemental figure 6b).

Steatosis-associated liver features were photographed at higher magnification (Figure 4e) and evaluated quantitatively (Figure 4f).

Minor issues

1. The authors state “INPP4B suppresses AKT and PKC signaling in liver by dephosphorylating PI(3,4)P2 and PI(4,5)P2 respectively thereby improving insulin sensitivity” (line 28). If the authors claim that the phenotype is dependent on PI(4,5)P2 then they need to show increased PI(4,5)P2 levels in INPP4B KO cells, as no studies have previously reported that INPP4B regulates cellular PI(4,5)P2 levels.

A: We thank the reviewer for this insightful comment. While INPP4B was shown to dephosphorylate PI(4,5)P2 *in vitro*, it was not demonstrated *in vivo*. We therefor deleted these statements from the manuscript discussion.

2. Several references are missing in introduction (line 41, line 43).

A: The references were added

3. INPP4B deletion should be validated by qPCR, western blot or IHC staining

A: We have recently published a manuscript in which *Inpp4b*^{-/-} mice were used. In that manuscript we provided INPP4B qPCR and IHC staining¹⁶. The INPP4B Western blot of the WT

and knockout animals was also reported previously by Ferron et al in supplemental figure 4e¹⁷. The mutation creates a frameshift in the reading frame and a stop codon in the N-terminus of the gene.

4. In Figure 1B and Supplementary table 1, it is not indicated whether there is a statistical difference in body weight between HFD fed WT and INPP4B-KO mice as described in the text (line 89).

A: In Figure 1b (currently Figure 1c) there are multiple groups that are significantly different from each other. This information is now added as a separate table (Supplemental table 1).

5. Leptin and adiponectin (line 106) should be in italics as this statement is referring to the genes and not the proteins.

A: Corrected

6. The authors state “The array analysis confirmed the increased leptin/adiponectin ratio at the protein level” (line 115). The mRNA data, which showed that only INPP4K-KO HFD mice had an increased lep/adipo ratio, does not match with the array analysis data, which showed that INPP4B-KO LFD, WT HFD and INPP4B-KO HFD mice all had an increased lep/adipo ratio to a similar level, and this should be acknowledged.

A: We have acknowledged this fact in the text of the manuscript.

7. The glucose and pyruvate tolerances tests in Figure 2a and Supplementary Figure 2a should also be quantified as area under the curve, and statistical comparisons between all groups should be performed.

A: We added AUCs to the glucose, pyruvate, and insulin tolerance tests.

8. Figure 5d-f are not described in the text.

A: We added description of these panel to the text.

9. The Ki67 staining in Supplementary Figure 5a does not appear to be altered in INPP4B WT versus KO HFD mice as the authors state (line 226), and quantification is needed to support this conclusion.

A: We have counted the number of Ki67 positive cells among prostate luminal epithelial cells. This graph is now included as a Supplemental figure 8b.

10. The authors state “Our data suggest that fasting-induced shut off of STEBP1 (typo) processing depends on functionally active INPP4B” (line 272). If the authors claim that INPP4B functional activity is

required for SREBP1 processing then they need to perform rescue experiments with expression of wild-type and catalytically inactive INPP4B.

A: We rephrased this statement to “Our data suggest that fasting-induced inhibition of SREBP1 processing is regulated by INPP4B”.

11. The authors state “Similar to our HFD Inpp4b^{-/-} mouse model, IL6 is elevated in the serum of patients with metastatic PCa” (line 285). As serum IL6 levels were not measured in mice, this experiment should be performed to confirm increased IL6 secretion.

A: As mentioned above, we have conducted serum measurements of inflammatory cytokines including IL-6 at 12 weeks of age and found no significant difference in serum IL-6 levels between 4 experimental groups. Therefore, we removed this statement.

12. Several quantifications do not match the relevant western blots (Figure 5g and 5i, Supplementary Figure 4a and 4b).

A: The referenced figure panels (Figures 6j-l and Supplemental figures 7a-b in the current version) were averaged from multiple western blots and individual bands on these blots: four for Figure 6j and six for Supplemental figure 7a. Thus, the exact distribution may appear somewhat different from the Western blot shown.

13. For western blots where multiple bands are present (Figure 5d and 5g), the band(s) of interest should be labeled.

A: The bands of interest are now labelled with an arrow.

14. Several graphs are missing labels or units (Figures 4a and 4b, Supplementary Figures 1b and 1c).

A: The labels are added

15. The depicted statistical comparisons in Supplementary Figure 5d and Figure 6k are not between the groups that the authors describe in the text. Statistical comparisons between WT and INPP4B-KO HFD mice should be depicted.

A: The values for statistical significance were added to figures 6d-f (former Figure 5d) and 7k (former Figure 6k).

Reviewer #2 (Remarks to the Author):

In the manuscript entitled “INPP4B protects from metabolic disease and associated disorders” by Zhang

and colleagues, the authors investigate the consequences of total ablation of the phosphatase INPP4B on metabolic disease and on prostatic intraepithelial neoplasia. While the basic phenotypes described, including increased body weight, fatty liver, altered GTT and presence of PIN in mice with deletion of this phosphatase are potentially interesting, the studies reported to characterize them physiologically and mechanistically appear still preliminary at this stage and require further in depth characterization.

1) The characterization of the fat phenotype of the KO in Fig 1 appears incomplete.

a) The authors should report the levels of INPP4B in all fat depots, not only of eWAT, to provide a clearer picture of the possible function of INPP4B in those depots.

b) The obesity phenotype in KO mice eluded to by the authors does not appear sufficiently supported by the differences shown in total body weight. Authors should provide information on fat and lean mass of WT and KO.

c) Weights of all fat depots, not just of eWAT of KO mice compared to WT, should be provided.

d) Complete pathological and morphological characterization of all fat depots by H and E staining should be presented.

e) Data on adipose cell size of WT and KO fat tissues to assess whether hyperplasia and/or hypertrophy is present in the KO should be added.

A: *Inpp4b* expression levels in brown adipose tissue and inguinal, epididymal, retroperitoneal, and mesenteric white adipose tissues was compared by quantitative RT-PCR. This data is presented in Figure 1b.

Fat and lean mass was measured by NMR and this data is shown in Figures 1d – f.

We have conducted H&E staining of inguinal, epididymal, retroperitoneal, and mesenteric WAT and BAT in four experimental groups. These images are shown in Supplementary figure 2. During histological evaluation we did not observed significant changes in the size of adipocytes, adipocyte hypertrophy or adipose tissue hyperplasia, and thus, we did not proceed to determine their size by flow cytometry. While excising fat depots for histology we observed that change in overall fat mass was proportional to increase in #4 mammary gland and epididymal WAT mass (Figures 1d, Supplemental figures 1d and 1e). Since all WAT depots were present in all experimental groups, we thought that measuring individual contributions to overall fat mass would provide few mechanistic insights into INPP4Bs role in metabolic function.

2) The authors claim that KO mice have increased inflammation in fat based on levels of some markers. Analysis of macrophage infiltration into fat tissues of WT and KO should be provided as a further evidence.

A: Adipose tissue in HFD *Inpp4b*^{-/-} males shows significantly higher levels of MCP-1 protein (Supplemental figure 3d) an important chemokine for macrophage recruitment¹⁸. Consistently, *Adgre1* (F4/80 antigen) a marker for populations of mouse tissue macrophages¹⁹ was significantly increased. Another marker, CD68, is routinely used as a marker of inflammation involving macrophages/monocytes and its levels are increased at both the mRNA and protein levels (Figures 2a, g, and h). We have also observed increased frequency of crown-like structures in WAT of the HFD *Inpp4b*^{-/-} males. Together this data strongly supports increased inflammation in WAT.

3) Graphs of the expression of different genes in the same tissue should be grouped in the same panel to make the results section less scattered and unfocused.

A: We agree with the reviewer that grouping different genes on the same graph would be more compact. However, we found that marking statistical significance between groups on such a graph would be difficult. To keep the data less scattered and focused we present it in the same consistent manner.

4) ITT should be performed and the effects of ablation of INPP4B on classic insulin signaling pathways should be described to definitively conclude impaired insulin sensitivity.

A: We performed the insulin tolerance test which showed that *Inpp4b*^{-/-} males are insulin resistant. The data is shown in Supplemental figure 4a.

5) The data provided on WT and KO livers are scarce. Further measurements to characterize the liver phenotype, including measurements of TG and quantification over multiple sections of liver biopsies of WT and KO of Mallory bodies (and other morphological changes reported) and of markers of inflammation (i. e: F4/80) should be provided.

A: As suggested by both reviewers, we provided high magnification images that feature the major steatotic features (Figure 4e). We also counted cells with microvesicular steatosis, macrovesicular steatosis, Mallory bodies, and pyknotic nuclei and presented this data as a bar graph in Figure panels 4f (individual features) and 4g (combined features). Of note, multiple features might be present in the same cells in livers of HFD *Inpp4b*^{-/-} males (Figure 4g).

6) The data on PIN is very preliminary. More detailed characterization of PIN should be provided.

A: We expanded our characterization of the prostate phenotype. We have counted the Ki67 positive cells to compare proliferation of the epithelial cells (Figure 8b). We stained prostates with E-cadherin to demonstrate the loss of polarity in epithelial cells (Supplemental figure 6a) and provided high magnification images of the atypical nuclei (Supplemental figure 6c) and stromal infiltration (Supplemental figure 6b).

7) The mechanistic explanation of the protective role of INPP4B in metabolism is tentative. The link between SREBP -PPARg the authors invoke is not clear.

A: Transgenic expression of the mature form of SREBP1 in mouse liver causes massive liver enlargement (3-4 fold by weight) and hepatosteatosis (12-21 fold increase in hepatic

triglycerides)²⁰. Accumulation of hepatic lipids inevitably leads to metabolic syndrome. We found that loss of INPP4B leads to an increase in levels of SREBP1 and causes an SREBP1 – associated hepatic phenotype and metabolic syndrome (Figures 3 and 4, Supplemental figure 5a). There is an extensive body of evidence that SREBP1 activates PPAR γ , either directly by induction of its expression²¹, or indirectly by stimulating production of PPAR γ activating ligands²². Consistently, in livers of *Inpp4b*^{-/-} males, we see increased expression of *Pparg* (Figure 5c) and increase in PPAR γ transcriptional activity (Figure 5e – i). Analysis of our RNAseq data confirmed activation of PPAR signaling pathways (Figure 5b). Thus, previously reported data and data obtained in our laboratory strongly suggests that INPP4B modulates the SREBP1-PPAR γ axis, and consequently, plays an important role in metabolic health.

- 1 Farmer, S. R. Transcriptional control of adipocyte formation. *Cell Metab* **4**, 263-273, doi:10.1016/j.cmet.2006.07.001 (2006).
- 2 Rosen, E. D. *et al.* C/EBP α induces adipogenesis through PPAR γ : a unified pathway. *Genes Dev* **16**, 22-26, doi:10.1101/gad.948702 (2002).
- 3 Mikkelsen, T. S. *et al.* Comparative epigenomic analysis of murine and human adipogenesis. *Cell* **143**, 156-169, doi:10.1016/j.cell.2010.09.006 (2010).
- 4 Clee, S. M. & Attie, A. D. The genetic landscape of type 2 diabetes in mice. *Endocr Rev* **28**, 48-83, doi:10.1210/er.2006-0035 (2007).
- 5 Taniguchi, C. M. *et al.* Divergent regulation of hepatic glucose and lipid metabolism by phosphoinositide 3-kinase via Akt and PKC λ /zeta. *Cell Metab* **3**, 343-353, doi:10.1016/j.cmet.2006.04.005 (2006).
- 6 Fleischmann, M. & Iynedjian, P. B. Regulation of sterol regulatory-element binding protein 1 gene expression in liver: role of insulin and protein kinase B/cAkt. *Biochem J* **349**, 13-17, doi:10.1042/0264-6021:3490013 (2000).
- 7 Porstmann, T. *et al.* SREBP activity is regulated by mTORC1 and contributes to Akt-dependent cell growth. *Cell Metab* **8**, 224-236, doi:10.1016/j.cmet.2008.07.007 (2008).
- 8 Hagiwara, A. *et al.* Hepatic mTORC2 activates glycolysis and lipogenesis through Akt, glucokinase, and SREBP1c. *Cell Metab* **15**, 725-738, doi:10.1016/j.cmet.2012.03.015 (2012).
- 9 Yecies, J. L. *et al.* Akt stimulates hepatic SREBP1c and lipogenesis through parallel mTORC1-dependent and independent pathways. *Cell Metab* **14**, 21-32, doi:10.1016/j.cmet.2011.06.002 (2011).
- 10 Sajan, M. P. *et al.* The critical role of atypical protein kinase C in activating hepatic SREBP-1c and NF κ B in obesity. *J Lipid Res* **50**, 1133-1145, doi:10.1194/jlr.M800520-JLR200 (2009).
- 11 Yamamoto, T. *et al.* Protein kinase C β mediates hepatic induction of sterol-regulatory element binding protein-1c by insulin. *J Lipid Res* **51**, 1859-1870, doi:10.1194/jlr.M004234 (2010).
- 12 Wu, J. & Dickson, A. J. SREBP isoform and SREBP target gene expression during rat primary hepatocyte culture. *In Vitro Cell Dev Biol Anim* **46**, 657-663, doi:10.1007/s11626-010-9321-3 (2010).
- 13 Wang, X. *et al.* E-cadherin bridges cell polarity and spindle orientation to ensure prostate epithelial integrity and prevent carcinogenesis in vivo. *PLoS Genet* **14**, e1007609, doi:10.1371/journal.pgen.1007609 (2018).
- 14 Ellwood-Yen, K. *et al.* Myc-driven murine prostate cancer shares molecular features with human prostate tumors. *Cancer Cell* **4**, 223-238, doi:10.1016/s1535-6108(03)00197-1 (2003).

- 15 Iwata, T. *et al.* MYC overexpression induces prostatic intraepithelial neoplasia and loss of Nkx3.1 in mouse luminal epithelial cells. *PLoS One* **5**, e9427, doi:10.1371/journal.pone.0009427 (2010).
- 16 Zhang, M. *et al.* Inositol polyphosphate 4-phosphatase type II regulation of androgen receptor activity. *Oncogene*, doi:10.1038/s41388-018-0498-3 (2018).
- 17 Ferron, M. *et al.* Inositol polyphosphate 4-phosphatase B as a regulator of bone mass in mice and humans. *Cell Metab* **14**, 466-477, doi:10.1016/j.cmet.2011.08.013 (2011).
- 18 Cranford, T. L. *et al.* Role of MCP-1 on inflammatory processes and metabolic dysfunction following high-fat feedings in the FVB/N strain. *Int J Obes (Lond)* **40**, 844-851, doi:10.1038/ijo.2015.244 (2016).
- 19 Schulz, C. *et al.* A lineage of myeloid cells independent of Myb and hematopoietic stem cells. *Science* **336**, 86-90, doi:10.1126/science.1219179 (2012).
- 20 Shimano, H. *et al.* Overproduction of cholesterol and fatty acids causes massive liver enlargement in transgenic mice expressing truncated SREBP-1a. *J Clin Invest* **98**, 1575-1584, doi:10.1172/JCI118951 (1996).
- 21 Fajas, L. *et al.* Regulation of peroxisome proliferator-activated receptor gamma expression by adipocyte differentiation and determination factor 1/sterol regulatory element binding protein 1: implications for adipocyte differentiation and metabolism. *Mol Cell Biol* **19**, 5495-5503, doi:10.1128/mcb.19.8.5495 (1999).
- 22 Kim, J. B., Wright, H. M., Wright, M. & Spiegelman, B. M. ADD1/SREBP1 activates PPARgamma through the production of endogenous ligand. *Proc Natl Acad Sci U S A* **95**, 4333-4337, doi:10.1073/pnas.95.8.4333 (1998).

Reviewers' comments:

Reviewer #1 (Remarks to the Author):

The authors have adequately addressed most of the comments, however some data requires further clarification which is listed below. Once these are amended, I would recommend this manuscript for publication.

1. The authors were asked to perform an insulin tolerance test which they have now included (Supplementary Figure 4I). Although this data shows that 4 month old INPP4B KO mice on LFD have significantly higher glucose levels, the WT mice did not exhibit a decrease in glucose levels as is expected following insulin administration (eg PMID: 23460866, Figure 3). This is most likely due to a technical error with the experiment, and this needs to be repeated as the current data suggests that these 4 month old WT LFD mice are also insulin resistant. Without showing a normal insulin response in WT mice, it cannot be concluded that INPP4B KO mice are insulin resistant.

2. Label the new Figures 6G-I with INPP4B KO to show these are rescue experiments.

Reviewer #2 (Remarks to the Author):

The authors adressed all the concerns of this reviewer.

Reviewer #1 (Remarks to the Author):

Q1. The authors were asked to perform an insulin tolerance test which they have now included (Supplementary Figure 4I). Although this data shows that 4 month old INPP4B KO mice on LFD have significantly higher glucose levels, the WT mice did not exhibit a decrease in glucose levels as is expected following insulin administration (eg PMID: 23460866, Figure 3). This is most likely due to a technical error with the experiment, and this needs to be repeated as the current data suggests that these 4 month old WT LFD mice are also insulin resistant. Without showing a normal insulin response in WT mice, it cannot be concluded that INPP4B KO mice are insulin resistant.

A: In the referenced manuscript PMID: 23460866, authors used C57/Bl6 while we used FVB mouse strain. Insulin signaling varies significantly between C57/Bl6 and FVB strains¹. Insulin administration caused 50 – 75 % decrease in glucose levels in C57/Bl6²⁻⁵ and only 20 – 40 % in FVB⁶⁻⁸.

We adjusted our insulin tolerance test parameters according to the protocol in the provided reference⁹ and observed that blood glucose levels in control group decreased 32.7 % one hour after insulin injection compared to the previous decrease of 17.9 %. The insulin tolerance profile for the control group is now within the reported response range for the FVB strain. Importantly, *Inpp4b*^{-/-} males remain insulin resistant under new conditions. The difference in AUC between WT and *Inpp4b*^{-/-} males increased in the new insulin tolerance test from 24 % to 34.6 %. We have replaced Supplemental figure 4a with the new insulin tolerance test data.

Q2. Label the new Figures 6G-I with INPP4B KO to show these are rescue experiments.

A. Figure Panels 6g-I now include *Inpp4b*^{-/-} label.

- 1 Berglund, E. D. *et al.* Glucose metabolism in vivo in four commonly used inbred mouse strains. *Diabetes* **57**, 1790-1799, doi:10.2337/db07-1615 (2008).
- 2 Pan, W. *et al.* Metabolic consequences of ENPP1 overexpression in adipose tissue. *Am J Physiol Endocrinol Metab* **301**, E901-911, doi:10.1152/ajpendo.00087.2011 (2011).
- 3 Shao, Y. *et al.* Effects of sleeve gastrectomy on the composition and diurnal oscillation of gut microbiota related to the metabolic improvements. *Surg Obes Relat Dis* **14**, 731-739, doi:10.1016/j.soard.2018.02.024 (2018).
- 4 Tsuneki, H. *et al.* Age-related insulin resistance in hypothalamus and peripheral tissues of orexin knockout mice. *Diabetologia* **51**, 657-667, doi:10.1007/s00125-008-0929-8 (2008).

- 5 Wong, C. K. *et al.* A high-fat diet rich in corn oil reduces spontaneous locomotor activity and induces insulin resistance in mice. *J Nutr Biochem* **26**, 319-326, doi:10.1016/j.jnutbio.2014.11.004 (2015).
- 6 de Pinho, L. *et al.* Diet composition modulates expression of sirtuins and renin-angiotensin system components in adipose tissue. *Obesity (Silver Spring)* **21**, 1830-1835, doi:10.1002/oby.20305 (2013).
- 7 Ludwig, D. S. *et al.* Melanin-concentrating hormone overexpression in transgenic mice leads to obesity and insulin resistance. *J Clin Invest* **107**, 379-386, doi:10.1172/JCI10660 (2001).
- 8 Wong, W. P. *et al.* Spontaneous diabetes in hemizygous human amylin transgenic mice that developed neither islet amyloid nor peripheral insulin resistance. *Diabetes* **57**, 2737-2744, doi:10.2337/db06-1755 (2008).
- 9 Roy, C. *et al.* Relationship of C5L2 receptor to skeletal muscle substrate utilization. *PLoS One* **8**, e57494, doi:10.1371/journal.pone.0057494 (2013).